# Data-Centric Defense: Shaping Loss Landscape with Augmentations to Counter Model Inversion

**Si Chen**                                                          *chensi@vt.edu*
*Virginia Tech*

**Feiyang Kang**                                                     *fyk@vt.edu*
*Virginia Tech*

**Nikhil Abhyankar**                                                 *nikhilsa@vt.edu*
*Virginia Tech*

**Ming Jin**                                                         *jinming@vt.edu*
*Virginia Tech*

**Ruoxi Jia**                                                        *ruoxijia@vt.edu*
*Virginia Tech*

**Reviewed on OpenReview:** *https://openreview.net/forum?id=r8wXaLJBIS*

## Abstract

Machine Learning models have shown susceptibility to various privacy attacks, with model inversion (MI) attacks posing a significant threat. Current defense techniques are mostly *model-centric*, involving modifying model training or inference. However, these approaches require model trainers' cooperation, are computationally expensive, and often result in a significant privacy-utility tradeoff. To address these limitations, we propose a novel *data-centric* approach to mitigate MI attacks. Compared to traditional model-centric techniques, our approach offers the unique advantage of enabling each individual user to control their data's privacy risk, aligning with findings from a Cisco survey that only a minority actively seek privacy protection. Specifically, we introduce several privacy-focused data augmentations that modify the private data uploaded to the model trainer. These augmentations shape the resulting model's loss landscape, making it challenging for attackers to generate private target samples. Additionally, we provide theoretical analysis to explain why such augmentations can reduce the risk of model inversion. We evaluate our approach against state-of-the-art MI attacks and demonstrate its effectiveness and robustness across various model architectures and datasets. Specifically, in standard face recognition benchmarks, we reduce face reconstruction success rates to $\leq 5\%$, while maintaining high utility with only a 2% classification accuracy drop, significantly surpassing state-of-the-art model-centric defenses. This is the first study to propose a data-centric approach for mitigating model inversion attacks, showing promising potential for decentralized privacy protection. Our code is available at https://github.com/SCccc21/DCD.git.

## 1 Introduction

Thanks to advances in computation and the availability of large-scale datasets collected globally, Machine Learning (ML) has experienced significant growth in recent years, showing great potential in various domains, such as computer vision, natural language processing, and healthcare, among others. However, the use of ML models trained on sensitive data can leak private information (Fredrikson et al., 2014; Shokri et al., 2017). While some data contributors may have a neutral stance or lack concern about their data privacy, a

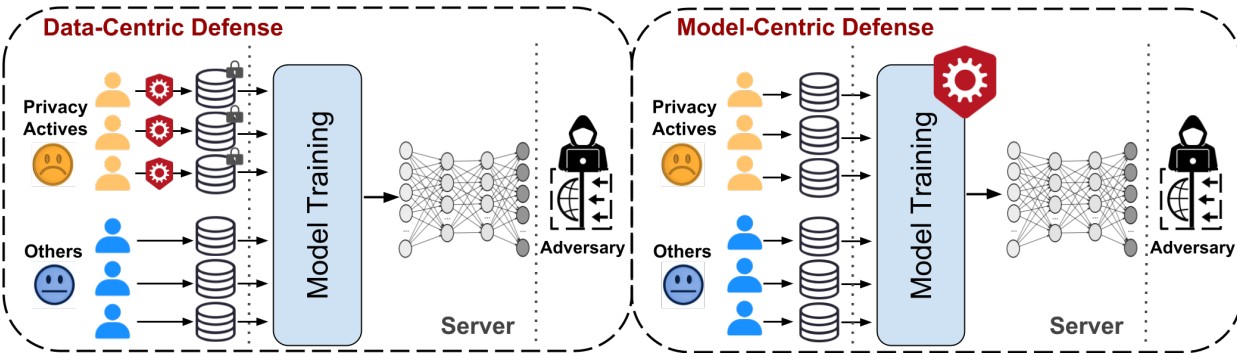

Figure 1: Data-Centric Defense vs. Model-Centric Defense.

survey conducted by Cisco (Cisco, 2022) in 2019 identified a group known as "privacy actives" who value and actively take steps to protect their privacy, including switching companies or providers. Moreover, existing legislations (e.g., the General Data Protection Regulation (GDPR) in the European Union (Magdziarczyk, 2019) and the California Consumer Privacy Act in the United States (Pardau, 2018)) stipulates the right of individual users to exercise control over their own data.

Existing defenses primarily focus on protecting privacy in a model-centric manner (Abadi et al., 2016; Jia et al., 2019; Wang et al., 2021; Yang et al., 2020), which involves altering model training (Abadi et al., 2016) or inference procedures (Jia et al., 2019). Common techniques include differentially private stochastic gradient descent (DP-SGD) (Abadi et al., 2016), which involves clipping and noising the gradients during training. These approaches often result in performance degradation and increased computation time. Moreover, they require users to trust the model trainer (e.g., the data-harvesting companies) to ensure privacy, limiting user control over their privacy risks. More critically, they often present a binary stance on privacy protection, offering protection to all users or none, overlooking the nuanced needs of individual users.

Real-world surveys Cisco (2022); Review (2020); Bongiovanni et al. (2020) reveal that only a small portion of users (i.e., 32%) are privacy actives, but the binary nature of existing solutions implies a significant compromise in utility for the sake of protecting the privacy of a minority. This motivates our exploration into *data-centric defenses*: *strategies that individuals can use to mitigate privacy attacks by modifying their data before uploading it to the central model trainer.* This empowers individuals to control their privacy risks in a decentralized manner. The randomized response (Warner, 1965), a long-standing strategy in social sciences, serves as an example, although it encounters challenges with high-dimensional data common in modern ML tasks.

In this paper, we focus on model inversion (MI) attacks to investigate the feasibility of effective data-centric defense. MI attacks, which reconstruct training data from a trained model, are well-researched and have been successful in both white-box and black-box scenarios (Fredrikson et al., 2014; Zhang et al., 2020b; Chen et al., 2021; Kahla et al., 2022; Struppek et al., 2022; An et al., 2022). Compared to other common privacy attacks such as membership inference attacks (Shokri et al., 2017; Nasr et al., 2019) (which infers whether certain data is used for training) and property inference attacks (Ganju et al., 2018; Melis et al., 2019; Song & Raghunathan, 2020) (which infers whether a dataset has certain global properties), MI attacks recover much more fine-grained information such as training images, posing a significant threat to user privacy. This work develops the first data-centric defense for MI attacks, making the following contributions:

①**MI Defense via Privacy-Focused Augmentations.** We propose privacy-focused data augmentations that can be injected by individual data contributors to mitigate their MI risks. Unlike traditional augmentations like cropping, rotation, and flipping that aim to improve model generalization, our augmentations are specifically tailored to thwart MI attacks. We present several ideas for designing such augmentations, with a central theme of shaping the loss landscape in ways that mislead MI attacks to recover irrelevant samples. This central theme distinguishes our ideas from the early simple randomized response, wherein the design of the noise injected into the data does not consider its impact on model behaviors. Also, in contrast

to existing MI defenses, our proposed approach, named DCD, requires no access to the victim model or training data from other contributors.

(2) **A Novel Privacy Protection Mechanism Setup.** Contrasting with the model-centric defense mechanisms that uniformly apply privacy protections to all users, we introduce a novel setup, inspired by insights from real-world surveys Cisco (2022); Review (2020); Bongiovanni et al. (2020). The novel setup is designed to cater specifically to the varying privacy needs of different users. It uniquely enables a selective privacy framework, where only a specific group of users — those who prioritize privacy according to their preferences and needs — engage in enhanced privacy measures. This innovative setup acknowledges and addresses the different privacy concerns in the user community, marking a significant shift from conventional binary protection setups that either extend privacy safeguards to all users or none.

(3) **Theoretical Analysis for Privacy-Focused Augmentations.** We provide theoretical justification for DCD, demonstrating that: 1) the proposed augmentations reshape the loss landscape near the target and inject irrelevant samples; 2) these treatments cause existing MI attacks relying on gradient-based optimization to converge to the irrelevant samples rather than the target samples.

(4) **Evaluation.** We evaluate DCD against various state-of-the-art MI attacks and demonstrate the robustness of DCD across different datasets, model architectures, and attack strategies. DCD outperforms the baselines by achieving a significantly improved privacy-utility tradeoff.

## 2  Background and Related Work

**Model Inversion Attacks.** In an MI attack, an adversary aims to reconstruct representative training samples for any target class of a victim model given access to the model. For example, in the context of face recognition, the adversary seeks to reconstruct face images of a specific target identity. To recover training data from a given model $f_\theta$ for any target class $y$, the key idea of MI is to find an input that minimizes the prediction loss of $y$: $x_{\text{syn}} \in \arg\min_x L(f_\theta(x), y)$.

However, solving this optimization over the high-dimensional space without any constraints generates noise-like features that lack semantic information and give unsatisfactory model inversion performance. Recently, GMI (Zhang et al., 2020b) proposed to optimize over the latent space of a pre-trained GAN instead: $x_{\text{syn}} = G(z^*)$, $z^* \in \arg\min_z L(f_\theta(G(z)), y) - D(G(z))$, where $G$ and $D$ represent the generator and the discriminator of the GAN, respectively. Chen et al. (2021); An et al. (2022); Struppek et al. (2022) follow the idea of using GAN and further improve the quality of reconstructed images with different techniques, e.g., knowledge distillation from the target model; latent space disentanglement via a StyleGAN (Karras et al., 2019; 2020a), etc. These works show that the samples synthesized by the GAN-based MI technique above can maintain high visual similarity to the original training data of $f_\theta$. The backbone of existing MI attacks involves solving an optimization objective, containing the prediction loss of the target class, i.e., $L(f_\theta(G(z)), y)$, and other quality-enhancing loss terms, via *gradient descent*. To recover multiple images, one could run gradient descent multiple times, each of which uses a randomly selected initialization value.

**Defense Techniques.** Existing defenses against MI involve altering the training process or model architectures. Differential privacy (DP) was deployed to defend MI in Fredrikson et al. (2014); Zhang et al. (2020b), which empirically show that DP can mitigate MI attacks only when the injected noise is large enough and as a side effect, the model suffers significant performance degradation. Wang et al. (2021) studied the theoretical basis of the inefficacy of DP in defending MI and introduced information bottleneck-based learning objectives to decrease the correlation between model outputs and training data. While improved over DP, it still suffers a significant privacy-utility tradeoff. Peng et al. (2022) proposed to minimize the dependency between the latent space and input while maximizing the dependency between the latent space and model outputs, enhancing utility. This, however, also requires modifications to model architectures. It's worth noting that all these defenses lack user control, relying on model trainers and imposing unnecessary utility sacrifices for privacy, especially when only a minority prioritize data privacy. In comparison, our approach involves only modification to data, which can be achieved by individual users who want to protect their privacy. Also, as we will show later, our defense effectively preserves the model's utility.

**Connection between Data Augmentation and Privacy.** The impact of augmentations on privacy risks has been studied recently in the context of membership inference attacks (Kaya & Dumitras, 2021; Tramèr et al., 2022; Chen et al., 2022). These attacks aim to determine if specific data samples were part of a model's training data. Kaya & Dumitras (2021) studied common data augmentations used for improving model generalization (e.g., random cropping and Gaussian augmentation) and empirically identified which ones mitigate or amplify membership inference risks. Tramèr et al. (2022); Chen et al. (2022) proposed augmenting the training set with mislabeled target samples to increase the risk. there is also a line of works that introduces corruption to decrease memorization for generative models (Karras et al., 2022; Daras et al., 2024b;a; Hans et al., 2024; Somepalli et al., 2023). While these works focus on improving diversity and performance in generative models, our paper addresses a distinct challenge: mitigating model inversion attacks in classification models, in which the impact of augmentations on privacy risks has not been explored. In addition to the difference in scope, our work distinguishes itself from existing research by going beyond the traditional collection of augmentations designed to improve model generalizability. Instead, we propose novel augmentations designed specifically to improve privacy, and such a design is grounded on an understanding of the influence of augmentations on the loss landscape of the victim model.

## 3 Methodology

**Notation and Setup.** Let $f_\theta$ denote a target victim classifier, which maps an input feature $x \in \mathcal{X}$ to a label $y \in \mathcal{Y}$, and $\mathcal{Y} = \{y_1, \ldots, y_k\}$. Denote the raw, unprotected training set by $\mathcal{D} = \{(x_{ij}, y_i) : i = 1, \ldots, k, j = 1, \ldots, m_i\}$, where $x_{ij}$ represents the $j$-th samples in class $i$ and $m_i$ is the total number of samples in class $i$. Take face recognition, a canonical application considered in the MI attack literature, as an example. Each $y_i$ represents a different identity or user, and $x_{ij}$ represents face images corresponding to identity $y_i$. Our goal is to protect training samples with the labels indexed by $S_{\text{tgt}}$ from model inversion attacks. This set will be referred to as the *target label set*. The raw training samples corresponding to the target label set can be represented as $D_{\text{tgt-raw}} = \{(x_{ij}, y_i) : i \in S_{\text{tgt}}, j = 1, \ldots, m_i\}$.

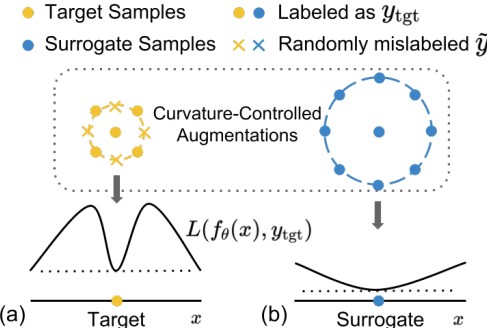

Figure 2: Illustration of curvature-controlled augmentations and the resulting loss landscape.

The primary assumption our method relies on is the fundamental optimization goal of MI attacks: to find an input x that minimizes the loss for a target class: $x_{\text{syn}} \in \arg\min_x L(f_\theta(x), y)$, where $y$ is the target class the attacker aims to recover, and $f_\theta$ is the victim model. We would like to note that this optimization goal forms the core principle underlying all existing MI attacks (An et al., 2022; Chen et al., 2021; Struppek et al., 2022; Kahla et al., 2022). Beyond this core assumption, our method is designed to be broadly applicable with minimal additional assumptions: 1) We don't make specific assumptions about the victim model architectures. 2) We don't assume any particular level of attacker access to the model (e.g., white-box, black-box). 3) We don't rely on assumptions about specific attack designs or algorithms.

### 3.1 Privacy-Focused Data Augmentations

Our approach introduces surrogate classes into the training set, designing augmentations to misdirect MI attacks toward recovering surrogate-class samples instead of target-class samples. We explain this process using a specific target class ($y_{\text{tgt}} \in \{y_i : i \in S_{\text{tgt}}\}$) for protection. When multiple target classes need protection (i.e., $|S_{\text{tgt}}| > 1$), one can easily apply the following process to each target class index in $S_{\text{tgt}}$.

**Surrogate Injection.** The process begins with identifying an "irrelevant" surrogate class ($y_{\text{srg}} \notin \mathcal{Y}$) for the target class ($y_{\text{tgt}}$), the reconstruction of which does not divulge sensitive information about the original target class. For example, in face recognition, a different public identity could serve as the surrogate class. We then gather samples from this surrogate class ($x_j^1, j = 1, \ldots, m, x_j^1 \sim P(X|y_{\text{srg}})$), *relabel* them as the target class, creating a mixed set of actual target and surrogate class samples labeled as the target class. The resulting augmented samples are denoted as $D_{y_{\text{tgt}}}^1 = \{(x_j^1, y_{\text{tgt}}) : j = 1, \ldots, m\}$.

The model trained *directly* on this mixture identifies both surrogate and target samples as the target class. Hence, an MI attack would generate a mix of target-class and surrogate-class samples. Detailed results are provided in Table 2. While this mix can obfuscate the adversary about the true attributes of the target class, our goal is to minimize the possibility of reconstructing the target class, thereby preventing the adversary from confidently determining the true attributes associated with the target class. The question now is *how to induce MI attacks to preferentially generate samples from the surrogate class over the target class.*

**Loss-Controlled Modification.** MI attacks essentially resolve optimization problems, seeking samples that result in the lowest loss when predicted as the target class. To counteract this, our first strategy modifies training data to slightly elevate the classification loss on the target compared to the surrogate, increasing the likelihood of detecting surrogate samples during MI optimization while reducing the chance for target samples. We accomplish this by randomly mislabeling a small fraction (denoted by $\pi_1$) of target samples, thereby increasing their loss, while leaving the surrogate samples' labels unaltered. The adjusted target samples are as follows:

$$D^0_{y_{\text{tgt}}} = \{(x^0_j, y'_j) : j = 1, ..., \lceil m\pi_1 \rceil\} \cup \{(x^0_j, y_{\text{tgt}}) : j = \lceil m\pi_1 \rceil + 1, ..., m\}, \tag{1}$$

where $x^0_j \sim P(X|y_{\text{tgt}})$ and $y'_j \sim \text{Uniform}(\mathcal{Y} \setminus y_{\text{tgt}})$.

**Curvature-Controlled Injection.** While loss-controlled modification consistently improves over direct surrogate injection, achieving nearly zero attack success rate, it can degrade model accuracy by 5% (details in Table 2). Leveraging the insight from non-convex optimization theory (Bertsekas, 1997), our second strategy manipulates the loss landscape's curvature, promoting a flatter curvature around surrogate samples and a steeper one near target samples. This approach biases the MI optimization towards reconstructing surrogate samples.

For surrogate samples, we employ Gaussian augmentations in their neighborhood, maintaining the same label, i.e., $D^2_{y_{\text{tgt}}} = (x^1_j + \mu_j, y_{\text{tgt}}) : j = 1, \ldots, m$, where $\mu_j \sim \mathcal{N}(0, \epsilon_1^2)$. This creates a flat loss landscape around surrogate samples. For target samples, we apply Gaussian augmentations but mislabel a portion of the augmented samples, denoted by $\pi_2$. The resulting augmented samples are:

$$D^3_{y_{\text{tgt}}} = \{(x^0_j + \mu'_j, \tilde{y}_j) : j = 1, \ldots, \lceil m\pi_2 \rceil\} \cup \{(x^0_j + \mu'_j, y_{\text{tgt}}) : j = \lceil m\pi_2 \rceil + 1, \ldots, m\}, \tag{2}$$

where $\mu'_j \sim \mathcal{N}(0, \epsilon_2^2)$ and $\tilde{y}_j \sim \text{Uniform}(\bar{\mathcal{Y}})$ where $\bar{\mathcal{Y}} \subset \{\mathcal{Y} \setminus y_{\text{tgt}}\}$ is some arbitrary subset. The trained model $f_\theta(\cdot)$, which tends to memorize training samples, will yield different label predictions for target samples and their close neighbors. This results in a large variation in $l(f_\theta(\cdot), y_{\text{tgt}})$ in the target samples' neighborhood (see Figure 2).

We refer to the complete injection process as **DCD**. The pseudocode is provided in Algorithm 1.[1]

**Choosing Hyperparameters.** In our main evaluation, we fix $\epsilon_1 = 8/255$, $\epsilon_2 = 0.003$, and $\pi_2 = 1$. Sensitivity analysis of defense performance to $\epsilon_2$, $\pi_1$, and $\pi_2$ are presented in Section 4.4.

## 3.2 Theoretical Analysis of Curvature-Controlled Injection

While it is relatively straightforward to see the impact of surrogate injection and loss control (i.e., injecting new minima and increasing the loss at the sensitive minima), understanding how curvature control manipulates the minima that gradient-based methods converge to is more nuanced. We demonstrate in this section of the paper that the proposed curvature control operations reshape the loss landscape around the target and surrogate samples. Leveraging the powerful *Capture Theorem*, we show that these treatments alter the convergence behavior of gradient-based optimization methods, redirecting them from the target samples

---

[1]Note that the injection increases the number of samples in the target class(es) by a factor of 4. While this could potentially signal malicious intent to remove privacy-focused augmentations, we assume the model trainer's honesty in this paper and leave concealing injected samples for future exploration. It's worth noting that real-world datasets (e.g., GTSRB (Stallkamp et al., 2011) used in our experiments) naturally have varying sample sizes across classes, which already poses challenges for intentional removal based solely on size.

to the surrogate samples. We establish conditions for the effectiveness of these techniques and provide a principled framework for their implementation.

**Curvature-Controlled Injection serves an implicit regularization on the eigenvalues of Hessian.** Consider neural networks constructed using continuous, piecewise affine activations (e.g., ReLU, leaky ReLU), we show that the correctly labeled Gaussian augmentations near surrogate samples will reduce the principal eigenvalue $\sigma_{\max}(\mathbf{H}_\varepsilon)$ of a Monte-Carlo approximation of $\varepsilon$-Hessian of loss (defined in (LeJeune et al., 2019)) near the surrogate (Lemma 1). Conversely, mislabeled Gaussian augmentations near target samples increase the principal eigenvalue near the target (Lemma 2).

**Lemma 1.** *Consider surrogate samples $D^1_{y_{tgt}} = \{(x^1_j, y_{tgt}) : j = 1, ..., m\}$ and the corresponding augmented set $D^2_{y_{tgt}} = \{(x^1_j + \mu_j, y_{tgt}) : j = 1, ..., m\}$. Then, compared to the loss function $\mathcal{L}$ of the model trained without noise augmentation $D^2_{y_{tgt}}$, the noise augmentation reduces the largest eigenvalue of a Monte-Carlo approximation of the Hessian matrix $\mathbf{H}_\varepsilon$ near the surrogate samples $D^1_{y_{tgt}}$ for the loss function of the model trained with $D^3_{y_{tgt}}$.*

**Lemma 2.** *Consider target samples with a mislabeling ratio $\pi_1$ given as $D^0_{y_{tgt}}$ defined in Eq. equation 1 and the corresponding augmented set with mislabeling $D^3_{y_{tgt}}$ defined in Eq. equation 2. Then, compared to the loss function $\mathcal{L}$ of the model trained without noise augmentation with mislabeling on $D^3_{y_{tgt}}$, the noise augmentation with mislabeling in $D^3_{y_{tgt}}$ increases the largest eigenvalue of a Monte-Carlo approximation of the Hessian matrix $\mathbf{H}_\varepsilon$ near the target samples $D^0_{y_{tgt}}$ for the loss function of the model trained with the noise-augmented set with mislabeling $D^3_{y_{tgt}}$.*

We defer formal lemma statements and proofs to Appendix B. Proof for Lemma 1 is a straightforward extension of that for Theorem 1 in LeJeune et al. (2019). Proof for Lemma 2 introduces a novel technique showing that minimizing the loss on noise-augmented samples with uniform mislabeling is ultimately equivalent to maximizing the loss on noise-augmented samples with correct labels, potentially of interest to the community studying the regularization effect of augmentations.

**Gradient-based optimization prefers flatter minima.** Let's now delve into how the previously outlined operations can influence the trajectory of gradient-based optimization. Specifically, they increase the likelihood of convergence towards surrogate samples while reducing for target samples. Capture Theorem (Bertsekas, 1997) states that the optimization trajectory tends to be attracted towards local optima once within sufficiently close proximity, given that the optimizer can converge. We'll outline the conditions that allow or prevent convergence of the gradient-based optimizer. Following that, we'll demonstrate how our loss-shaping operations directly impact these conditions, thereby theoretically guiding the optimization trajectory to favor convergence at surrogate samples. The subsequent theorem provides a formal explanation for the termination of gradient-based nonlinear optimization when using a constant stepsize—a method extensively utilized in current MI attacks (Zhang et al., 2020b; Struppek et al., 2022; Chen et al., 2021). While our analysis isn't limited to constant stepsizes, we'll postpone the discussion on variable stepsizes to the Appendix.

**Theorem 1 (Convergence of gradient method (Bertsekas, 1997)).** *Let $\{x^k\}$ be a sequence generated by a gradient method $x^{k+1} = x^k + \alpha^k d^k$, where $\{d^k\}$ is gradient related. Assume that the gradient of $f$ is $L$-Lipschitz, and that for all $k$ we have $d^k \neq 0$ and*

$$\epsilon \leq \alpha^k \leq (2 - \epsilon)\bar{\alpha}^k, \quad where \quad \bar{\alpha}^k = \frac{|\nabla f(x^k)' d^k|}{L\|d^k\|^2}$$

*and $\epsilon \in (0, 1]$ is a fixed scalar. Then every limit point of $\{x^k\}$ is a stationary point of $f$.*

**Remark 1 (Lipschitz of loss gradients directly affects convergence at local optima).** *Theorem 1 asserts that a gradient-based optimizer converges to a local optimum if the stepsize lies within a certain range. This range's upper limit is inversely proportional to the Lipschitz constant of the loss gradient in the area, and convergence will fail if the stepsize exceeds this range. In essence, local optima with larger Lipschitz constants require smaller step sizes for convergence, while those with smaller Lipschitz constants can accommodate a broader range of stepsizes.*

**Remark 2 (Reshaping convergence through noise-augmentation and mislabeling).** *The Lipschitz constant of the loss gradient in a region equals the largest eigenvalue of the loss Hessian, $\sigma_{\max}(\mathbf{H})$. Increasing $\sigma_{\max}(\mathbf{H})$ in a local optimum's capture region (as in Lemma 1) rejects convergence for optimizers with non-minimal stepsizes. Conversely, decreasing $\sigma_{\max}(\mathbf{H})$ (as in Lemma 2) accommodates a wider range of stepsizes. However, excessively small stepsizes may be practically infeasible due to inevitable noises from gradient partial estimation and round-off/quantization errors. Also, for nonconvex loss functions typical in neural networks, optimization with extremely small stepsizes is generally impractical and results in poor performance. Thus, the proposed loss landscape shaping essentially lowers the likelihood of convergence at target samples for gradient-based optimizers, steering the optimization trajectory toward surrogate samples.*

**Remark 3 (Elevating loss with mislabeling strengthens effects).** *Finally, augmenting noise and mislabeling samples near target samples to elevate loss creates barriers on the loss landscape. These barriers prevent gradient-based optimizers from entering the capture region of target samples, especially those with smaller stepsizes. The optimizer's trajectory is diverted early to avoid loss increase before reaching the barrier's ridge, which contradicts the requirement for smaller stepsizes. Consequently, it becomes less likely for gradient-based optimizers to reach and converge at the target samples' capture region.*

## 4 Experiments

In this section, we first evaluate the effectiveness of various defense methods when protecting different number of targets in defending GMI, a classic MI attack (Section 4.2). We show that DCD is effective across various number of targets, but yields the best performance especially when protecting a small amount of targets. Then, we aim to answer several key questions and provide a comprehensive understanding of the strengths and weaknesses of DCD: 1) How does DCD compare to existing MI defenses in terms of model utility and robustness to various MI attacks in ? 2) Does DCD work well across datasets and model architectures? 3) How to choose hyperparameters for DCD? 4) How to choose surrogate samples? Through our evaluation, we seek to provide insights and empirical evidence that shed light on the aforementioned questions and contribute to a comprehensive understanding of the strengths of DCD. An overview of the experimental setups is provided in Table 6.

### 4.1 Setup

**Attack Algorithms.** We assess the effectiveness of our defense against four MI attacks in white-box setting: GMI[2] (Zhang et al., 2020b), PPA[3] (Struppek et al., 2022), MIRROR-W[4] (An et al., 2022), and PLG-MI[5] (Yuan et al., 2023). GMI is the most classic MI attack in the literature, while PPA, MIRROR-W, and PLG-MI represent the most recent ones achieving state-of-the-art attack performance. For completeness, we also evaluate our defense against the most recent black-box attack, MIRROR-B, though it has been shown less potent than the white-box counterpart. We utilize open-sourced implementations of these attacks and faithfully replicate their settings.

**Datasets and Models.** We demonstrate the efficacy of DCD across multiple tasks and datasets that are commonly employed in previous studies on MI attacks (Zhang et al., 2020b; Struppek et al., 2022; An et al., 2022; Chen et al., 2021): (1) Traffic Sign Recognition (GTSRB (Stallkamp et al., 2011)); (2) Face Recognition (CelebA (Liu et al., 2015), FaceScrub (Ng & Winkler, 2014)); and (3) Dog Classification (St.Dogs (Khosla et al., 2011)). We evaluate our defense on various target models with different architectures including VGG-16(Simonyan & Zisserman, 2014), ResNeSt-101(Zhang et al., 2020a), ResNet-152(He et al., 2016), ResNext-101(Xie et al., 2017), and DenseNet-169(Huang et al., 2017). Following the setup in the original attack algorithms, we use GANs pre-trained on public datasets from domains similar to the private datasets used to train target models. Table 6 provides an overview of the datasets and models utilized in our experiments.

---

[2] https://github.com/SCccc21/Knowledge-Enriched-DMI
[3] https://github.com/LukasStruppek/Plug-and-Play-Attacks/tree/master
[4] https://github.com/njuaplusplus/mirror
[5] https://github.com/LetheSec/PLG-MI-Attack

**Baselines.** We compare DCD with DP-SGD (Abadi et al., 2016), MID (Wang et al., 2021) and BiDO (Peng et al., 2022). To ensure consistent evaluation, we utilized their open-source implementations (Wang, 2021; Peng, 2022). We carefully select the privacy parameters by testing various configurations of each baseline. These parameters include privacy budget, noise multiplier, and gradient clipping threshold for DP; weight of information loss for MID; weights of the two dependency loss $\lambda_x$ and $\lambda_y$ for BiDO. Detailed information on the hyperparameter selection is available in Table C.4. We would like to emphasize that DP-SGD is very time-consuming: it increases the training time by 8.94 when training a face recognition model on CelebA. By contrast, our proposed algorithm, DCD, ensures a comparable training time to the original method (approximately 1.02 times).

**Evaluation Protocol.** We conducted a comprehensive evaluation of our defense mechanism under this new setup, focusing on both utility and privacy metrics. In terms of utility, we measure the classification accuracy of the target model on the entire clean test set (**ACC-all**) and the target test set (**ACC-tar**) which only consists of images from target class that need to be protected. For privacy, we evaluate the attack accuracy (**Att. ACC**), which corresponds to the classification accuracy of an evaluation model on inverted samples. Evaluation models are trained using different architectures from the target models following Zhang et al. (2020b); Chen et al. (2021); Struppek et al. (2022). For the GMI attack, we generate 500 samples for each target class and average the results across 5 target classes. For PPA and MIRROR attacks, we generate 50 samples for each target, averaging over 10 targets for PPA and 8 targets for MIRROR. For PLG-MI, we generate 50 samples for each target, averaging over 300 targets. These target classes are *randomly* selected. To examine the effectiveness of our method when protecting a larger portion of users, we performed a sensitivity analysis on the number of protected targets (detailed in Section 4.2). Specifically, we explore scenarios with 10%, 50%, and 100% of users are privacy actives, allowing us to assess our method's adaptability to varying degrees of user privacy concerns.

**Implementation of DCD.** In the experiments, we fixed $\epsilon_1 = 8/255$, $\epsilon_2 = 0.003$, and $\pi_2 = 1$. We use $\pi_1 = 0.2$ for GMI, MIRROR, and PLG-MI, $\pi_1 = 0.3$ for PPA. Regarding surrogate selection, for the datasets GTSRB, FaceScrub, and St.Dogs, we *randomly* selected surrogate classes from within each dataset. The target models are then trained on the remaining classes. For CelebA, the target models are trained on the top 1,000 identities based on the sample quantity, with surrogates *randomly* selected from the remaining. For VGGFace2, since it is no longer available publicly, we are only able to collect 8 classes for training the target model, with a surrogate randomly chosen from CelebA protecting all. The guideline for automated surrogate selection is provided in Section 4.4, with the code provided in the supplementary materials.

We implemented DCD to defend against the existing MI Attacks for multiple models and datasets in Python 3.9.12 using PyTorch version 1.12.1. The experiments were carried out on one server having eight NVIDIA RTX A6000 GPUs with CUDA 12.1.

### 4.2 Performance Evaluation when Protecting Different Number of Targets.

Real-world motivation drives us to assess defense performance in scenarios where only a portion of users is deeply concerned about their privacy. In this section, we use GMI, one of the most classic MI attacks as an example, to study DCD's capabilities in protecting different portion of target classes. We follow the standard setup in (An et al., 2022; Chen et al., 2021), where the target classifiers are trained on 1,000 identities from CelebA with the most number of samples. Then, we randomly select surrogates samples from the remaining identities. We vary the number of targets for protection (i.e., 10, 500, 1000) and evaluate the defense performance of all defense methods.

As depicted in Figure 5, DCD consistently achieved the lowest attack accuracy and demonstrated a significant advantage in preserving model utilities, especially when protecting a small portion of targets ($\leq 50\%$). In contrast, model-centric baselines [6] exhibited higher variance in attack accuracy when protecting different targets. In the case of safeguarding all of the training targets, DCD's accuracy was only slightly lower

---

[6]DP-SGD (Abadi et al., 2016) is commonly utilized to apply differential privacy, employing a *uniform* privacy parameter across all data points. While recent research (Li et al., 2017; Heo et al., 2023) has introduced mechanisms that assign person-

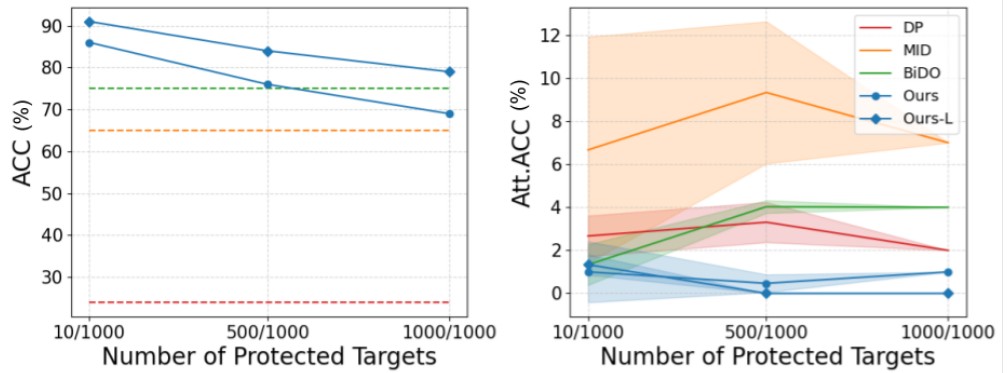

Figure 3: Defense performance against GMI on CelebA dataset. Ours-L denotes the use of DCD with a larger model (i.e., ResNet-152), whereas Ours, MID, BiDO, and DP are trained with VGG-16. The attack results are averaged over three runs, each with randomly selected protected targets. Notably, DCD yields the best privacy-utility tradeoffs when protecting a small amount of targets (i.e., ≤ 50%), and remains effective across various protection ratio.

than the most advanced model-centric defense method, BiDO. This marginal difference could potentially be addressed by adopting a larger model capacity - indicated as Ours-L in Figure 5, which represents our method with a larger model (i.e., IR-152). This leads to the highest accuracy compared to all other baselines, with the attack accuracy remaining consistently low, below 1%. Implementing a larger model is also a practical option when using DCD. In practical terms, service providers adopting our strategy can judiciously select models, gravitating towards larger architectures that exhibit heightened resilience to the label noise introduced by our defense. Notably, with the amplification in model size, DP, MID and BiDO suffer a larger privacy-utility tradeoff. Consequently, they lack the leverage to utilize increased model dimensions for attenuating this tradeoff, a feat achievable by our data-centric method.

In our main evaluation, we focus on the setting where only a small portion of privacy actives, closely mirroring real-world scenario; we show that as a data-centric defense, DCD provides flexible privacy controls and achieves near-zero privacy-utility tradeoff under this setting, outperforms existing model-centric defenses.

### 4.3 Main Results

**Comparison with Model-Centric Baselines.** We compare DCD with the previous state-of-the-art defenses on various MI attacks, datasets, and model architectures under the novel setup. *To better understand the performance when using different surrogates*, the results of DCD in Table 1 are averaged over three runs, where each run uses a different set of surrogates that are randomly selected.

As shown in the table, DCD outperforms the baselines in both utility and privacy metrics. The unprotected models exhibit alarmingly high attack accuracy, with GMI at 76%, PPA at 90%, MIRROR at 100%, and PLG-MI at 89%. In contrast, DCD significantly reduces the attack accuracy to 0% for GMI, MIRROR, and PLG-MI attacks, and to 1.55% for PPA. This suggests the robustness of DCD

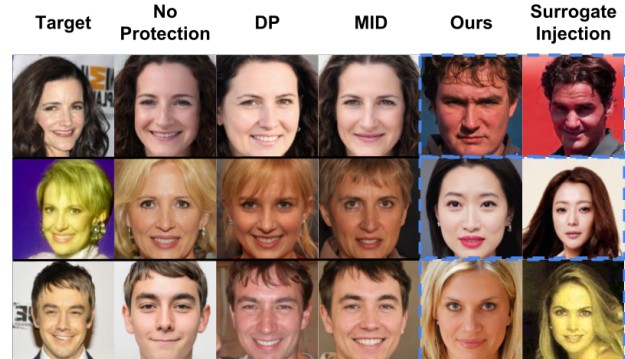

Figure 4: Visual comparison of MI recovered face samples with different defenses. Each row shows reconstructions of the same identity under different defenses, with true images on the left and our surrogate injection on the right.

alized privacy parameters to individual data points, there is an absence of open-source implementations for these personalized approaches. In this paper, we focus on the comparison to Abadi et al. (2016).

against attack algorithms. A notable advantage of
DCD is its ability to balance privacy and utility well. Unlike model-centric baselines, which exhibit a sub-stantial drop in classification accuracy, our method ensures high classification accuracy, with a decrease of less than 3% on the face datasets CelebA and VGGFace2. We also include the evaluation of DCD on a most recent black-box attacks BREP-MI (Kahla et al., 2022) in Appendix D.

To qualitatively examine the defense results, we plot some samples generated by PPA when deploying different defense methods in Figure 4. As shown in the last two rows, DCD successfully fools MI into generating samples resembling the surrogate ones. Other methods, however, continue to produce samples that closely mirroring the actual protected targets, where sensitive information such as the gender and hair color can still be leaked.

Table 1: Defense performance comparison against state-of-the-art MI attacks. Results are given in %, with symbols ↑ and ↓ respectively indicate that higher and lower scores give better defense performance. Note that for MIRROR, all classes are target classes, and the classification accuracy is demoted as ACC. Additionally, DCD results are averaged over three runs, each with a different surrogate selection, and the standard deviations are provided in the table. The minimal variance underscores the robustness of DCD to the choice of surrogate selection.

| | GMI TSRD→GTSRB | | | PPA FFHQ→CelebA | | | MIRROR-W FFHQ→VGGFace2 | | MIRROR-B FFHQ→VGGFace2 | | PLG-MI FFHQ→CelebA | | |
|---|---|---|---|---|---|---|---|---|---|---|---|---|---|
| | ACC-all↑ | ACC-tar↑ | Att. ACC↓ | ACC-all↑ | ACC-tar↑ | Att. ACC↓ | ACC↑ | Att. ACC↓ | ACC↑ | Att. ACC↓ | ACC-all↑ | ACC-tar↑ | Att. ACC↓ |
| No Protection | 98.34 | 99.20 | 76.13 | 88.42 | 84.37 | 90.40 | 99.99 | 100.0 | 99.99 | 100.0 | 88.02 | 88.99 | 89.40 |
| DP | 54.30 | 31.24 | 12.80 | 39.61 | 6.67 | 14.33 | 56.25 | 54.69 | 56.25 | 50.00 | 24.47 | 25.56 | 64.09 |
| MID | 67.70 | 55.37 | 54.53 | 69.54 | 53.33 | 52.33 | 41.34 | 100.00 | 41.34 | 12.50 | 74.77 | 73.56 | 87.12 |
| BIDO | 87.02 | 72.62 | 54.40 | 74.92 | 50.00 | 19.33 | 83.66 | 89.06 | 83.66 | 87.50 | 75.33 | 75.40 | 4.03 |
| DCD | 96.21± 0.58 | 93.25±0.89 | 0.00± 0.00 | 87.67±0.54 | 80.41±1.28 | 1.55±0.79 | 96.88±0.20 | 0.00±0.00 | 96.88±0.20 | 0.00±0.00 | 77.90±0.38 | 74.86±1.12 | 0.00±0.00 |

**Generalization to Different Datasets.** We further evaluate the performance of DCD on different datasets, focusing on one of the most advanced MI attacks, PPA (Struppek et al., 2022). Our evalua-tion considers three datasets: CelebA, FaceScrub, and St.Dogs; and we employ StyleGAN2 that have been pre-trained on public datasets with different distributional shifts (Karras et al., 2020b). Consistent with the previous findings, Table 9 shows that DCD achieves an impressive privacy-utility tradeoff, effectively reducing the attack accuracy to <5% on all datasets while causing a minimal impact on the model accuracy of the target class. The accuracy remains high for all datasets with only a slight drop that < 1%.

**Generalization to Different Model Architectures.** Furthermore, we thoroughly evaluate the perfor-mance of DCD across a range of popular model architectures, including ResNest, ResNet, ResNext, and DenseNet. The results, as shown in Table 8, highlight the robustness of our method across different choices of architectures used during model training. Notably, DCD consistently reduces the attack accuracy to be less than 5% across all models, even when the initial attack accuracy is as high as 96%. As a data-centric defense, DCD does not require access to training procedures or the choice of model architectures. It effec-tively protects privacy by focusing on the data itself, ensuring that sensitive information remains secure and independent of specific modeling decisions.

## 4.4 Analysis and Ablations

We proposed a couple of ideas in Section 3 to improve our defense performance, including 1) surrogate injec-tion (Surr-Inj), 2) loss control (L-Ctrl), and 3) curvature control (C-Ctrl). We now present a comprehensive analysis of each choice point of our approach.

**Ablation Study on Each Design Idea.** We have shown that the combination of all these ideas can lead to significant defense performance improvement over model-centric baselines. Here, we conduct an ablation study to investigate the improvement introduced by each individual idea and the hyperparameters. Table 2 presents the results of protecting a target class in the GTSRB dataset against GMI attacks. We observe that solely injecting surrogate samples in the training set does not effectively mitigate the risk of MI attacks. However, when combined with either loss control or curvature control, the attack accuracy decreases to approximately 10%. By employing all three techniques together, we reduce attack accuracy to 0.

Table 2: Ablation Study of ideas in DCD. $\pi_1$ only involved in Loss Control (L-Ctrl) and $\pi_2$ only involved in Curvature Control (C-Ctrl). Larger mislabel ratios can result in lower attack accuracy but also lower clean accuracy. We show that the combination of Loss Control and Curvature Control yields the best privacy-utility tradeoff.

| | No Protection | Surr-Inj | Surr-Inj&L-Ctrl | | | Surr-Inj&C-Ctrl | | | | Surr-Inj&L-Ctrl&C-Ctrl | | |
|---|---|---|---|---|---|---|---|---|---|---|---|---|
| Mislabel Ratio $\pi_1$ | - | - | 0.1 | 0.2 | 0.5 | - | - | - | - | 0.1 | 0.2 | 0.2 |
| Mislabel Ratio $\pi_2$ | - | - | - | - | - | 0.1 | 0.2 | 0.5 | 1 | 0.5 | 0.5 | 1 |
| ACC-all ↑ | 98.58 | 98.46 | 98.14 | 97.98 | 97.89 | 98.50 | 98.62 | 97.87 | 97.86 | 98.39 | 97.97 | 97.96 |
| ACC-tar ↑ | 99.25 | 100.00 | 98.45 | 97.97 | 95.15 | 99.42 | 99.71 | 98.55 | 98.51 | 98.99 | 97.94 | 97.38 |
| Att. ACC ↓ | 79.20 | 29.60 | 12.60 | 9.80 | 0.60 | 21.80 | 19.80 | 11.80 | 10.60 | 0.30 | 0.00 | 0.00 |

**Sensitive Analysis on Noise Magnitude of Target Samples.** In addition to analyzing the mislabel ratio for loss control and curvature control in Table 2, we conduct a supplementary experiment to investigate the influence of different noise magnitudes on target samples $\epsilon_2$. It is important to note that, throughout this paper, we maintain a fixed noise magnitude of $\epsilon_1 = 8/255$ for all experiments. By selecting $\epsilon_2$ values that are smaller than $\epsilon_1$, we can further enhance the control strength and create sharper curvature in the target samples. As expected, the results in Table 3 demonstrate that DCD achieves comparable and satisfactory performance when using $\epsilon_2 < 8/255$, with the best performance observed at $\epsilon_2 = 0.003$. On the other hand, for $\epsilon_2 > 8/255$, the strength of curvature control weakens, resulting in a lower defense performance (i.e.,Att.ACC around 30%).

Table 3: Sensitive analysis on the noise magnitude of target samples $\epsilon_2$. Experiments are conducted on GTSRB with GMI attack. Injected samples use a magnitude of 8/255. Note that mislabel ratios are set to be $\pi_1 = 0$, $\pi_2 = 0.5$ to amplify the effect brought by $\epsilon_2$. Using a $\epsilon_2 \leq 8/255$ can achieve good performance.

| | Gaussian Noise Magnitude $\epsilon_2$ | | | | | | |
|---|---|---|---|---|---|---|---|
| | 0.001 | 0.003 | 0.005 | 0.01 | 8/255 | 0.1 | 0.3 |
| ACC-all↑ | 97.75 | 97.21 | 98.16 | 97.458 | 98.12 | 97.32 | 97.32 |
| ACC-tar↑ | 99.13 | 98.99 | 95.57 | 99.71 | 99.13 | 99.86 | 99.57 |
| Att. ACC↓ | 2.60 | 0.40 | 2.00 | 2.20 | 5.80 | 26.20 | 35.40 |

Table 4: DCD's defense performance with full mismatch and full match surrogate samples, where the selection of surrogate samples that have different attribute (i.e., Gender, Hair Color) leads to better defense performance.

| Attribute | | Defense Performance | | | | | | |
|---|---|---|---|---|---|---|---|---|
| Gender | Hair Color | | ACC | Att. ACC | | ACC(−−) | Att. ACC(−−) | | ACC(++) | Att. ACC(++) |
| Male | Black | | 83.33 | 96.77 | | 81.67 | 0.00 | | 100.00 | 5.99 |
| Female | Black | | 100.00 | 100.00 | | 100.00 | 4.99 | | 100.00 | 47.99 |
| Female | Blonde | | 85.71 | 92.00 | | 81.14 | 0.00 | | 85.71 | 7.99 |
| Male | Blonde | | 75.00 | 100.00 | | 69.00 | 0.00 | | 75.00 | 0.00 |

**How to Choose Surrogate Samples?** We investigate the impact of using different surrogate samples and provide a guideline to choose them properly. Specifically, we found that there are two desiderata for conducting a more successful defense:

*A. Less similarity between surrogate and target samples.*
We observe that using surrogate samples that differ significantly from the target can enhance defense performance. This is because such injection would result in the recovery of images that appear very different from

Table 5: Impact of diversity and quality of surrogate samples within the same class.

|  | No Protection | Dup-1-Low | Dup-1-High | Dup-5 | No-Dup |
|---|---|---|---|---|---|
| **ACC-all**↑ | 86.95 | 86.92 | 86.24 | 86.97 | 86.57 |
| **ACC-tar**↑ | 100.00 | 96.47 | 97.13 | 97.52 | 97.15 |
| **Att. ACC**↓ | 100.00 | 22.50 | 18.00 | 0.80 | 4.50 |

the target images. For instance, when targeting a male with black hair, we collect images from a female with blonde hair as our surrogate. For an in-depth investigation, we conduct an experiment on a face recognition model trained on 1,000 identities with the most number of samples from the CelebA dataset. We focus on attributes like gender and hair color which are predominantly identifiable, and randomly select four target identities with varying combinations of gender and hair color attributes. For each target, we choose two surrogate identities from the remaining dataset outside the 1,000 training classes: one is a full mismatch (marked as '−−') with distinct gender and hair color, and the other is a full match (marked as '++') sharing the target's gender and hair color.

As shown in Table 4, a full match ('++') can reduce the attack accuracy to <10% for three out of the four target identities. However, one identity (female with black hair) exhibits a relatively high attack accuracy of 47.99%. This discrepancy may be attributed to the higher vulnerability of this particular target to MI attacks, as it has a significantly high attack accuracy of 100% without any protection. Since a full-match surrogate shares identical attributes with the target, the risk of potential recovery of sensitive attributes still exists. In contrast, a full mismatch('−−') successfully reduces the attack accuracy of all target identities to <10%, with three identities achieving a perfect defense (0% attack accuracy), aligning with our expectations. This demonstrates that employing surrogate samples that significantly differ from the target samples can yield superior defense performance.

*B. Small but non-zero diversity among surrogate samples within the same class.*

Selecting surrogate samples from public celebrities is one of the most convenient ways to collect surrogates which a large number of diverse samples are available online, and it is important to understand the impact of quality and diversity of surrogate samples on the defense performance. We focus on four target classes with an initial high attack accuracy of 100% without protection, and evaluate in three scenarios where the same amount of surrogates are collected: 1) No-Dup: each surrogate image is unique; 2) Dup-5: 5 diverse surrogate images are collected for each target and duplicated; 3) Dup-1: a single image is collected for each target and duplicated. The sample in Dup-1 scenario is selected from the five collected samples in the Dup-5 scenario, with one being of high quality (Dup-1-High) and another of low quality (Dup-1-Low) based on visual factors such as occlusion of the face by hair or other elements that may impact the overall image quality.

Table 5 demonstrates that all three scenarios maintain high utility. In terms of privacy, No-Dup yields an attack accuracy of 4.5% on these vulnerable targets. By using less diverse surrogate samples (Dup-5), the defense performance is further improved, resulting in an attack accuracy of 0.8%. We also observe that the presence of diversity among surrogate samples is crucial, as purely duplicated surrogate samples lead to a relatively higher attack accuracy. Besides, using high-quality surrogate samples leads to lower attack accuracy compared with low-quality ones. One possible explanation is that the target model fails to learn well about the low-quality surrogate samples with partial occlusion, thereby weakening the effectiveness of our proposed loss control mechanism.

## 5 Conclusion

Our paper introduces the first user-empowered, data-centric defense mechanism, DCD, for mitigating data privacy risks. Supported by theoretical analysis and extensive evaluations, DCD effectively counters model inversion attacks and surpasses model-centric baselines in utility and privacy. It does, however, increase the number of samples in target classes, potentially alerting malicious model trainers. Future work aims to obscure these injected samples to address this concern.

## 6 Limitations and Discussion

The introduction of surrogate samples into the target class means these surrogates will be classified as belonging to the target class, posing a potential security risk. It is also crucial to note that this risk is confined strictly to the user represented by the target class. That is, while surrogate identities introduced can bypass the face recognition system and gain access, they can only do so for that specific target class. Moreover, the selection of these surrogates rests entirely in the hands of the user represented by the target class. Given that publicizing their surrogate samples would endanger their own security, a logical user would not be motivated to disclose this information. As a result, we believe the likelihood of an adversary discerning and exploiting a user's specific surrogate samples remains minimal in practice; therefore, the associated security risk is also minimal.

We also note that irrespective of the protective measures in place and the specific defense strategy employed, MI attack techniques can pose inherent security risks. Malicious attackers can exploit existing MI attack techniques to recover samples identified as the target class. When used maliciously, these samples could potentially provide unauthorized access related to that target class, especially if the model serves such functions. However, samples recovered through MI might be readily detected by the operator of the targeted machine learning system. For instance, MI attacks mostly rely on pre-trained GANs to generate samples; such samples typically exhibit certain high-frequency artifacts not found in natural samples, as detailed in Frank et al. (2020). Such MI-generated samples could potentially be detected through straightforward frequency analysis. Addressing the broader security implications of general MI attacks goes beyond the purview of this paper, and we aim to explore this in-depth in future research.

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

# A    Pseudo-code

---

**Algorithm 1:** Algorithm of DCD.

---

**Input** : Entire label set $\mathcal{Y}$, target label set $S_{tgt}$, raw training samples corresponding to the target label set $D_{\text{tgt-raw}}$, mislabel ratio $\pi_1$ and $\pi_2$, noise magnitude $\epsilon_1$ and $\epsilon_2$.

**1** Denote samples from class $y_i$ as $\{(x_{ij}, y_i) : j = 1, \ldots, m_i\}$, where $m_i$ is the number of samples of this class.

**2 for** $i \in S_{tgt}$ **do**

**3**   Find a surrogate class not present in $\mathcal{Y}$ and gather the same number of samples as class $y_i$. Relabel the gathered samples as class $y_i$: $D_i^1 = \{(x_{ij}^1, y_i) : j = 1, ..., m_i\}$.

**4**   Mislabel a small portion of raw target training samples with a ratio $\pi_1$ using a random wrong label $y' \sim \text{Uniform}(\mathcal{Y} \setminus y_i)$ to these samples:
$D_i^0 = \{(x_{ij}^0, y_j') : j = 1, ..., \lceil m_i\pi_1 \rceil\} \cup \{(x_{ij}^0, y_i) : j = \lceil m_i\pi_1 \rceil + 1, ..., m_i\}$.

**5**   Augment surrogate samples with Gaussian noise: $D_i^2 = \{(x_{ij}^1 + \mu_j, y_i) : j = 1, \ldots, m_i\}$, where $\mu_j \sim \mathcal{N}(0, \epsilon_1^2)$.

**6**   Augment target samples with Gaussian noise, and mislabel a portion of augmentations with ratio $\pi_2$ using random wrong label $\tilde{y}$:
$D_i^3 = \{(x_{ij}^0 + \mu_j', \tilde{y}_j) : j = 1, \ldots, \lceil m_i\pi_2 \rceil\} \cup \{(x_{ij}^0 + \mu_j', y_i) : j = \lceil m_i\pi_2 \rceil + 1, \ldots, m_i\}$, where $\mu_j' \sim \mathcal{N}(0, \epsilon_2^2)$.

**7 end**

**8 return** $\{D_i^0 \cup D_i^1 \cup D_i^2 \cup D_i^3 : i \in S_{tgt}\}$

---

# B    Proofs

## B.1    Formal statement of Lemma 1 and proof

**Lemma 1 (formal).** *Consider a deep network constructed using continuous, piecewise affine activations (e.g., ReLU) as defined in (LeJeune et al., 2019). let $f(\mathbf{x})$ represent the mapping from the input to the output, which partitions the input space $\mathbb{R}^D$ based on the activation patterns. Within such a vector quantization (VQ) region of the network, $f$ is simply an affine mapping that can be written as a continuous, piecewise affine operator $f(\mathbf{x}) = \mathbf{A}[\mathbf{x}]\mathbf{x} + \mathbf{b}[\mathbf{x}]$. Assume the loss function $\mathcal{L}$ is $L$-Lipschitz. Consider surrogate samples $D_{y_{tgt}}^1 = \{(x_j^1, y_{tgt}), j = 1, ..., m\}$ and the corresponding augmented set $D_{y_{tgt}}^2 = \{(x_j^1 + \mu_j, y_{tgt}) : j = 1, ..., m\}$. Then, the loss on the augmented samples $\mathcal{L}^{aug}$ can be bounded by*

$$\mathcal{L}^{aug} \leq \mathcal{L}^{sur} + L \cdot \left[ \|x_j^1\| \cdot \|\mathbf{A}[x_j^1 + \epsilon_1 \cdot \mu_j] - \mathbf{A}[x_j^1]\|_2 + \|\mathbf{b}[x_j^1 + \epsilon_1 \cdot \mu_j] - \mathbf{b}[x_j^1]\|_2 + \delta \cdot \|\mathbf{A}[x_j^1 + \epsilon_1 \cdot \mu_j]\|_2 \right]$$

*where $\mathcal{L}^{sur}$ denotes the loss on surrogate samples, and $\|\mathbf{A}[x_j^1 + \epsilon_1 \cdot \mu_j] - \mathbf{A}[x_j^1]\|_2$ is a Monte Carlo approximation to the spectral norm of $\varepsilon$-approximation of Hessian of the loss function $\mathbf{H}_\varepsilon$ near the surrogate samples $D_{y_{tgt}}^1$, which bounds its largest eigenvalue as $\frac{1}{\epsilon_1} \|\mathbf{A}[x_j^1 + \epsilon_1 \cdot \mu_j] - \mathbf{A}[x_j^1]\|_2 = \sigma_{\max}(\mathbf{H}_\varepsilon)$*

*Proof.* As defined in (LeJeune et al., 2019), let $f(\mathbf{x})$ represent the mapping from the input to the output of a deep network constructed using continuous, piecewise affine activations (e.g., ReLU), which partitions the input space $\mathbb{R}^D$ based on the activation patterns. Within such a *vector quantization (VQ)* region of the network, $f$ is simply an affine mapping that can be written as a continuous, piecewise affine operator

$$f(\mathbf{x}) = \mathbf{A}[\mathbf{x}]\mathbf{x} + \mathbf{b}[\mathbf{x}]$$

Then, consider the model loss $\mathcal{L}$ on a sample $(x_j^1 + \epsilon_1 \cdot \mu_j, y)$ from the noise-augmented set $D_y^2$, we have

$$
\begin{aligned}
\mathcal{L}[f(x_j^1 + \epsilon_1 \cdot \mu_j), y] =& \mathcal{L}[\mathbf{A}[x_j^1 + \epsilon_1 \cdot \mu_j](x_j^1 + \epsilon_1 \cdot \mu_j) + \mathbf{b}[x_j^1 + \epsilon_1 \cdot \mu_j], y] \\
=& \mathcal{L}[\mathbf{A}[x_j^1]x_j^1 + \mathbf{b}[x_j^1]x_j^1 + (\mathbf{A}[x_j^1 + \epsilon_1 \cdot \mu_j] - \mathbf{A}[x_j^1])x_j^1 \\
& + \mathbf{b}[x_j^1 + \epsilon_1 \cdot \mu_j] - \mathbf{b}[x_j^1] + \mathbf{A}[x_j^1 + \epsilon_1 \cdot \mu_j]\epsilon_1 \cdot \mu_j, y] \\
=& \mathcal{L}[\mathbf{A}[x_j^1]x_j^1 + \mathbf{b}[x_j^1]x_j^1, y] + \big[ (\mathbf{A}[x_j^1 + \epsilon_1 \cdot \mu_j] - \mathbf{A}[x_j^1])x_j^1 \\
& + \mathbf{b}[x_j^1 + \epsilon_1 \cdot \mu_j] - \mathbf{b}[x_j^1] + \mathbf{A}[x_j^1 + \epsilon_1 \cdot \mu_j]\epsilon_1 \cdot \mu_j \big]^T \nabla_f \mathcal{L}[f(x_j^1), y] \\
& + \text{h.o.t.}
\end{aligned}
$$

where the last equation performs a Taylor expansion. Assume the loss function $\mathcal{L}$ is $L$-Lipschitz in this region. For some scalar $\delta > 0$ that $\|\epsilon_1 \cdot \mu_j\| \le \delta$ holds with high probability, we have

$$
\begin{aligned}
\mathcal{L}[f(x_j^1 + \epsilon_1 \cdot \mu_j), y] \approx& \mathcal{L}[\mathbf{A}[x_j^1]x_j^1 + \mathbf{b}[x_j^1]x_j^1, y] + \big[ (\mathbf{A}[x_j^1 + \epsilon_1 \cdot \mu_j] - \mathbf{A}[x_j^1])x_j^1 \\
& + \mathbf{b}[x_j^1 + \epsilon_1 \cdot \mu_j] - \mathbf{b}[x_j^1] + \mathbf{A}[x_j^1 + \epsilon_1 \cdot \mu_j]\epsilon_1 \cdot \mu_j \big]^T \nabla_f \mathcal{L}[f(x_j^1), y] \\
\le& \mathcal{L}[f(x_j^1), y] + L \cdot \big[ \|x_j^1\| \cdot \|\mathbf{A}[x_j^1 + \epsilon_1 \cdot \mu_j] - \mathbf{A}[x_j^1]\|_2 \\
& + \|\mathbf{b}[x_j^1 + \epsilon_1 \cdot \mu_j] - \mathbf{b}[x_j^1]\|_2 + \delta \cdot \|\mathbf{A}[x_j^1 + \epsilon_1 \cdot \mu_j]\|_2 \big]
\end{aligned}
\tag{3}
$$

where $\| \cdot \|_2$ denotes the spectral norm, which is equal to the largest eigenvalue $\| \cdot \|_2 = \sigma_{\max}(\cdot)$.

Using the notions from (LeJeune et al., 2019), we extend the definition of Hessian for neural network models with piecewise affine activations (e.g., ReLU). Let $\varepsilon > 0$, for $\mathbf{x}$ where the loss function $\mathbf{x}$ is differentiable and an arbitrary unit vector $\mathbf{u}$, we define $\varepsilon$-approximation of Hessian as

$$
\mathbf{H}_\varepsilon[\mathbf{u}] := \frac{1}{\varepsilon}(\mathbf{A}[\mathbf{x} + \varepsilon\mathbf{u}] - \mathbf{A}[\mathbf{x}])
\tag{4}
$$

which is consistent with the finite element definition of the Hessian and recovers the Hessian as $\varepsilon \to 0$. Thus, $\|\mathbf{A}[x_j^1 + \epsilon_1 \cdot \mu_j] - \mathbf{A}[x_j^1]\|_2$ in Eq. equation 3 is a Monte Carlo approximation ((LeJeune et al., 2019)) to the spectral norm of $\varepsilon$-approximation of Hessian of the loss function $\mathbf{H}_\varepsilon \to \nabla_f^2 \mathcal{L}(\cdot, \cdot)$ near the surrogate samples $D_{y_{\text{tgt}}}^1$, which bounds its largest eigenvalue as $\frac{1}{\epsilon_1}\|\mathbf{A}[x_j^1 + \epsilon_1 \cdot \mu_j] - \mathbf{A}[x_j^1]\|_2 = \sigma_{\max}(\mathbf{H}_\varepsilon)$. Minimizing the loss on samples $(x_j^1 + \epsilon_1 \cdot \mu_j, y)$ from the noise-augmented set $D_{y_{\text{tgt}}}^2$ reduces the upper bound on the largest eigenvalue of a Monte-Carlo approximation to the $\varepsilon$-approximation of Hessian $\mathbf{H}_\varepsilon$ of the loss function $\sigma_{\max}(\mathbf{H}_\varepsilon)$ near the surrogate samples $D_{y_{\text{tgt}}}^1$.

**Q.E.D.** $\hfill\square$

### B.2 Formal statement of Lemma 2 and proof

**Lemma 2 (formal).** *Consider a deep network constructed using continuous, piecewise affine activations (e.g., ReLU) as defined in (LeJeune et al., 2019). let $f(\mathbf{x})$ represent the mapping from the input to the output, which partitions the input space $\mathbb{R}^D$ based on the activation patterns. Within such a vector quantization (VQ) region of the network, $f$ is simply an affine mapping that can be written as a continuous, piecewise affine operator $f(\mathbf{x}) = \mathbf{A}[\mathbf{x}]\mathbf{x} + \mathbf{b}[\mathbf{x}]$. Assume the loss function $\mathcal{L}$ is $L$-Lipschitz. Consider target samples with a mislabeling ratio $\pi_1$ given as $D_{y_{tgt}}^0$ defined in Eq. equation 1 and the corresponding augmented set with mislabeling $D_{y_{tgt}}^3$ defined in Eq. equation 2. Then, the expected loss on the augmented samples $\mathcal{L}^{aug}$ can be bounded by*

$$
\mathbb{E}_{y' \sim Uniform\{\bar{\mathcal{Y}} \subset \{\mathcal{Y} \backslash y\}\}} \mathcal{L}^{aug} \ge -\frac{1}{k-1} \cdot \log\big(1 - g_y[f(x_j^0 + \epsilon_2 \cdot \mu_j)]\big)
$$

*where $g(\cdot)$ denotes the Softmax function in the classification loss defined as $g_y[f(x_j^0 + \epsilon_2 \cdot \mu_j)] = \frac{\exp[f_y(x_j^0 + \epsilon_2 \cdot \mu_j)]}{\sum_{y_k \in \mathcal{Y}} \exp[f_{y_k}(x_j^0 + \epsilon_2 \cdot \mu_j)]}$ with the loss on target samples $\mathcal{L}[f(x_j^0 + \epsilon_2 \cdot \mu_j), y] = -\log g_y[f(x_j^0 + \epsilon_2 \cdot \mu_j)]$ bounded in Lemma 1 and $k = |\bar{\mathcal{Y}}|$.*

*Proof.* Consider the model loss $\mathcal{L}$ on a sample from the noise-augmented set with uniform mislabeling $D^3_{y_{\mathrm{tgt}}} = \{(x_j^0 + \epsilon_2 \cdot \mu_j, y') : j = 1, ..., m_1\} \cup \{(x_j^0 + \epsilon_2 \cdot \mu_j, y) : j = m_1 + 1, ..., m\}$, we have

$$\mathbb{E}_{y' \sim \mathrm{Uniform}\{\bar{\mathcal{Y}} \subset \{\mathcal{Y} \setminus y\}\}} \mathcal{L}[f(x_j^0 + \epsilon_2 \cdot \mu_j), y'] = \frac{1}{k-1} \sum_{y_i \in \{\mathcal{Y} \setminus y\}} \mathcal{L}[f(x_j^0 + \epsilon_2 \cdot \mu_j), y_i] \tag{5}$$

where we define $k = |\bar{\mathcal{Y}}|$ as the total number of wrong labels in $\bar{\mathcal{Y}}$. Consider typical cross-entropy classification loss with Softmax given as Foundation (Retrieved May 13, 2023, from https://pytorch.org/docs/stable/generated/torch.nn.CrossEntropyLoss.html)

$$\mathcal{L}[f(x_j^0 + \epsilon_2 \cdot \mu_j), y] = -\log \frac{\exp[f_y(x_j^0 + \epsilon_2 \cdot \mu_j)]}{\sum_{y_k \in \mathcal{Y}} \exp[f_{y_k}(x_j^0 + \epsilon_2 \cdot \mu_j)]}$$

for noise-augmented samples with correct labels and

$$\mathcal{L}[f(x_j^0 + \epsilon_2 \cdot \mu_j), y'] = -\log \frac{\exp[f_{y'}(x_j^0 + \epsilon_2 \cdot \mu_j)]}{\sum_{y_k \in \mathcal{Y}} \exp[f_{y_k}(x_j^0 + \epsilon_2 \cdot \mu_j)]}$$

for noise-augmented samples with uniform mislabeling. Let $g(\cdot)$ denote the Softmax function in the classification loss–that is

$$g_y[f(x_j^0 + \epsilon_2 \cdot \mu_j)] = \frac{\exp[f_y(x_j^0 + \epsilon_2 \cdot \mu_j)]}{\sum_{y_k \in \mathcal{Y}} \exp[f_{y_k}(x_j^0 + \epsilon_2 \cdot \mu_j)]}, \ g_{y'}[f(x_j^0 + \epsilon_2 \cdot \mu_j)] = \frac{\exp[f_{y'}(x_j^0 + \epsilon_2 \cdot \mu_j)]}{\sum_{y_k \in \mathcal{Y}} \exp[f_{y_k}(x_j^0 + \epsilon_2 \cdot \mu_j)]}$$

where $g_y[f(x_j^0 + \epsilon_2 \cdot \mu_j)]$ and $g_{y'}[f(x_j^0 + \epsilon_2 \cdot \mu_j)]$ denotes the Softmax function of classification loss for noise-augmented samples with correct labels and with uniform mislabeling, respectively. Naturally, we have $g_y[f(x_j^0 + \epsilon_2 \cdot \mu_j)] + \sum_{y_i \in \bar{\mathcal{Y}}} g_{y_i}[f(x_j^0 + \epsilon_2 \cdot \mu_j)] = 1$.

Then, for Eq. equation 5, we have

$$\begin{aligned} \mathbb{E}_{y' \sim \mathrm{Uniform}\{\mathcal{Y} \setminus y\}} \mathcal{L}[f(x_j^0 + \epsilon_2 \cdot \mu_j), y'] &= \frac{1}{k-1} \sum_{y_i \in \bar{\mathcal{Y}}} -\log g_{y_i}[f(x_j^0 + \epsilon_2 \cdot \mu_j)] \\ &= -\frac{1}{k-1} \cdot \log \prod_{y_i \in \bar{\mathcal{Y}}} g_{y_i}[f(x_j^0 + \epsilon_2 \cdot \mu_j)] \\ &\geq -\frac{1}{k-1} \cdot \log \sum_{y_i \in \bar{\mathcal{Y}}} g_{y_i}[f(x_j^0 + \epsilon_2 \cdot \mu_j)] \\ &= -\frac{1}{k-1} \cdot \log \left(1 - g_y[f(x_j^0 + \epsilon_2 \cdot \mu_j)]\right) \geq 0 \end{aligned} \tag{6}$$

where the inequality is based on the AM–GM inequality (Hirschhorn, 2007). Eq. equation 6 states that minimizing the loss on noise-augmented samples with uniform mislabeling will minimize the upper bounds on the negation of $\log \left(1 - g_y[f(x_j^0 + \epsilon_2 \cdot \mu_j)]\right)$, which is equivalent to maximizing the lower bounds on $\log \left(1 - g_y[f(x_j^0 + \epsilon_2 \cdot \mu_j)]\right)$. This equals to maximizing the quantity $1 - g_y[f(x_j^0 + \epsilon_2 \cdot \mu_j)]$, which is equal to minimizing $g_y[f(x_j^0 + \epsilon_2 \cdot \mu_j)]$. Given that the loss on noise-augmented samples with correct labels is given as $\mathcal{L}[f(x_j^0 + \epsilon_2 \cdot \mu_j), y] = -\log g_y[f(x_j^0 + \epsilon_2 \cdot \mu_j)]$, this means minimizing the loss on noise-augmented samples with uniform mislabeling is ultimately equivalent to maximizing the loss on noise-augmented samples with correct labels.

Note that Lemma 1 has shown that the model loss on noise-augmented samples with correct labels upper bounds the Monte-Carlo approximation to the spectral norm of $\varepsilon$-approximation of Hessian $\mathbf{H}_\varepsilon$ of loss function, which upper bounds the largest eigenvalue of Monte-Carlo approximation to the $\varepsilon$-approximation of Hessian $\sigma_{\max}(\mathbf{H})$ near the target samples $D^0_{y_{\mathrm{tgt}}}$. Thus, minimizing the loss on samples from the noise-augmented set with uniform mislabeling $D^3_{y_{\mathrm{tgt}}} = \{(x_j^0 + \epsilon_2 \cdot \mu_j, y') : j = 1, ..., m_1\} \cup \{(x_j^0 + \epsilon_2 \cdot \mu_j, y) : j = m_1 + 1, ..., m\}$, equivalent to maximizing the loss on samples with the same noise-augmentation but correct labels, increases the upper bound on the largest eigenvalue of a Monte-Carlo approximation to the $\varepsilon$-approximation of Hessian $\mathbf{H}_\varepsilon$ of loss function near the target samples $D^0_{y_{\mathrm{tgt}}}$.

**Q.E.D.** □

### B.3 Other Theorems

**Theorem 2 (Capture Theorem (restated, ([Bertsekas, 1997]))).** *Let $f$ be continuously differentiable and let $\{x^k\}$ be a sequence satisfying $f(x^{k+1}) \leq f(x^k)$ for all $k$ and generated by a gradient method $x^{k+1} = x^k + \alpha^k d^k$, which is convergent in the sense that every limit point of sequences that it generates is a stationary point of $f$. Assume that there exist scalars $s > 0$ and $c > 0$ such that for all $k$ there holds*

$$\alpha^k \leq s, \quad \|d^k\| \leq c\|\nabla f(x^k)\|$$

*Let $x^*$ be a local optimum of $f$, which is the only stationary point of $f$ within some open set. Then there exists an open set $S$ containing $x^*$ such that if $x^{\bar{k}} \in S$ for some $\bar{k} \geq 0$, then $x^k \in S$ for all $k \geq \bar{k}$ and $\{x^k\} \to x^*$. Furthermore, given any scalar $\bar{\epsilon} > 0$, the set $S$ can be chosen so that $\|x - x^*\| < \bar{\epsilon}$ for all $x \in S$*

*Proof.* See ([Bertsekas, 1997]). $\qquad\square$

**Theorem 3 (Convergence of gradient method – constant stepsize (restated, ([Bertsekas, 1997]))).** *Let $\{x^k\}$ be a sequence generated by a gradient method $x^{k+1} = x^k + \alpha^k d^k$, where $\{d^k\}$ is gradient related. Assume that the gradient of $f$ is $L$-Lipschitz, and that for all $k$ we have $d^k \neq 0$ and*

$$\epsilon \leq \alpha^k \leq (2 - \epsilon)\bar{\alpha}^k,$$

*where*

$$\bar{\alpha}^k = \frac{|\nabla f(x^k)'d^k|}{L\|d^k\|^2},$$

*and $\epsilon \in (0, 1]$ is a fixed scalar. Then every limit point of $\{x^k\}$ is a stationary point of $f$.*

*Proof.* See ([Bertsekas, 1997]). $\qquad\square$

## C Experimental Details

In this section, we discuss the details of our experimental setup for code reproducibility.

### C.1 Hardware and Software Details

We implemented DCD to defend against the existing MI Attacks for multiple models and datasets in Python 3.9.12 using PyTorch version 1.12.1. The experiments were carried out on one server having eight NVIDIA RTX A6000 GPUs with CUDA 12.1.

### C.2 Datasets

**CelebA**   A large-scale dataset consisting of 202,599 images of 10,177 different celebrities of the size 178x218. We further crop the images by a face factor of 0.65 [7] and resize the images to 224x224. We are using the 1000 most frequent celebrity faces (identities with the most number of samples) as a part of our dataset which constitutes of 27,034 training samples and 3,004 test samples. The dataset is available at https://mmlab.ie.cuhk.edu.hk/projects/CelebA.html.

**FaceScrub**   The FaceScrub is also a large-scale face dataset comprising 106,863 face images belonging to 530 celebrities (265 male and 265 female) with each celebrity having roughly 200 images. We mapped the images such that the integers 0-264 belong to male celebrities and 265-529 represent female celebrities. We follow the settings in PPA ([Struppek et al., 2022]) to use 34,090 training images and 3,788 test images. The dataset is available at http://vintage.winklerbros.net/facescrub.html.

---

[7] https://github.com/LynnHo/HD-CelebA-Cropper

**VGGFace2**   The VGGFace2 is a large-scale face recognition dataset, in which images are downloaded from Google Image Search and have large variations in pose, age, illumination, ethnicity and profession. Since the dataset link is no longer active on the official website [8], we are only able to collect 1984 training images and 416 test images belonging to 8 different classes.

**Stanford Dogs**   The Stanford Dogs is a dog classification dataset having 120 dog breeds represented in 18,522 training and 2,058 test samples, summing up to a total of 20,580 images. The images vary in their sizes, styles, and content with a few images also containing multiple dog breeds. The dataset is available at http://vision.stanford.edu/aditya86/ImageNetDogs/.

**GTSRB**   GTSRB or German Traffic Sign Recognition Benchmark is a traffic signal recognition dataset having 35,288 training images and 12,630 test images all belonging to 43 distinct classes. The images are resized to 32x32. The dataset is available at https://benchmark.ini.rub.de/.

**Flickr-Faces-HQ (FFHQ)**   FFHQ is a highly diverse and high-quality dataset (better than CelebA and FaceScrub) with 70,000 face images of resolution 1024x1024. The dataset is available at https://github.com/NVlabs/ffhq-dataset.

**MetFaces**   A 1,336-strong image dataset having varied artistic versions of human faces. The dataset is however biased and contains a limited representation of people with darker skins. The dataset is available at https://github.com/NVlabs/metfaces-dataset.

**Animal Faces-HQ (AFHQ)**   The dataset contains 512x512 sized 16,130 images of wildlife animals, cats, and dogs. Since the dataset is used for the evaluation of Stanford Dogs, we select only the images of dogs. The dataset is available at https://github.com/clovaai/stargan-v2.

**TSRD**   It is a collection of 58 categories including 6164 traffic sign images. The training and test images are split into 4170 images and 1994 images respectively. The dataset is available at https://opendatalab.com/TSRD.

### C.3   Attack Implementation Details

We discuss various attacks and the methodologies to evaluate DCD. In our experiments, We assess the effectiveness of our defense against four MI attacks in white-box setting: GMI[9] (Zhang et al., 2020b), PPA[10] (Struppek et al., 2022), MIRROR-W[11] (An et al., 2022), and PLG-MI[12] (Yuan et al., 2023). GMI is the most classic MI attack method in the literature, while PPA, MIRROR-W, and PLG-MI represent the most recent ones achieving state-of-the-art attack performance.

For completeness, we also evaluate our defense against the most recent black-box attacks, MIRROR-B and BREP-MI [13]Kahla et al. (2022), though they have been shown less potent than the white-box counterpart.

We utilize open-sourced implementations of these attacks and faithfully replicate their settings in our experiments.

### C.4   Baseline Implementation Details

This section provides the implementation details of the two baselines used to compare DCD with. DP-SGD involves adding noise to the gradient and gradient clipping. The hyperparameters include the probability upper bound, denoted as $\delta$, which represents the likelihood of the model failing to provide privacy guarantees (roughly $\frac{1}{size\ of\ the\ dataset}$), and the noise multiplier, denoted as $\sigma$, which is adjusted to achieve the desired

---

[8]https://www.robots.ox.ac.uk/~vgg/data/vgg_face2/

[9]https://github.com/SCccc21/Knowledge-Enriched-DMI

[10]https://github.com/LukasStruppek/Plug-and-Play-Attacks/tree/master

[11]https://github.com/njuaplusplus/mirror

[12]https://github.com/LetheSec/PLG-MI-Attack

[13]https://github.com/m-kahla/Label-Only-Model-Inversion-Attacks-via-Boundary-Repulsion/tree/main

Table 6: Overview of the attack methods, datasets, and models on which DCD is evaluated. Note that for BREP-MI and PLG-MI, the GAN is trained on a subset of data from CelebA, which is disjoint from the private part.

| Attack Method | Task | Private Dataset | Public Dataset | Pre-trained GAN | Model |
|---|---|---|---|---|---|
| GMI | Traffic Sign Recognition | GTSRB | TSRD | WGAN | VGG-16 |
| PPA | Face Recognition | CelebA | FFHQ | StyleGAN2[14] | ResNeSt-101, ResNet-152, ResNext-101, DenseNet-169 |
| | | | MetFaces | | ResNeSt-101 |
| | | FaceScrub | FFHQ | StyleGAN2 | ResNeSt-101 |
| | | | MetFaces | | ResNeSt-101 |
| | Dog Classification | St.Dogs | AFHQ | StyleGAN2 | ResNeSt-101 |
| MIRROR-W | Face Recognition | CelebA-partial256 | VGGFace2 | StyleGAN[15] | VGG-16 |
| MIRROR-B | Face Recognition | CelebA-partial256 | VGGFace2 | StyleGAN | VGG-16 |
| PLG-MI | Face Recognition | CelebA | CelebA | WGAN[16] | VGG-16 |
| BREP-MI | Face Recognition | CelebA | CelebA | WGAN | face.evoLVe, IR-152 |

privacy budget $\epsilon$. The learning rate and batch size remain fixed at the values used for normal model training, while the threshold for gradient clipping is set to a constant value of 1.

The goal of MID is to restrict the information conveyed by the model's prediction about the input. To achieve this, MID introduces a hyperparameter denoted as $\beta$, which represents the weight assigned to the information loss that reduces the correlation between the output logit and the input. Detailed information is provided in Table 7.

BiDO proposes two additional loss terms: one to minimize the dependency between input data and hidden representations, while the other to maximize the dependency between hidden representations and model outputs. The two loss terms are controled by hyperparameters $\lambda_x$ and $\lambda_y$ respectively. Intuitively, larger $\lambda_x$ results in lower dependency between input data and hidden representations, which helps prevent privacy leakage; and larger $\lambda_y$ results in higher dependency between hidden and model outputs, which helps preserve model utility. We follow the guideline from the paper to choose $\lambda_x$ and $\lambda_y$ that maximize privacy while minimizing utility loss.

Table 7: Privacy Parameters in DP-SGD, MID and BIDO.

| Attack Method | MID | DP | | | BIDO | |
|---|---|---|---|---|---|---|
| | $\beta$ | $\sigma$ | $\delta$ | $C$ | $\lambda_x$ | $\lambda_y$ |
| **GMI** | 0.2 | 1.0 | $1e-4$ | 1.0 | 1.0 | 0.7 |
| **PPA** | 0.07 | 0.1 | $4e-5$ | 1.0 | 0.05 | 0.1 |
| **MIRROR** | 0.003 | 2.0 | $5e-4$ | 1.0 | 4.0 | 20.0 |
| **PLG-MI** | 0.02 | 0.01 | $4e-5$ | 1.0 | 0.1 | 2.0 |

# D   Additional Evaluation Results

**Generalization to Different Model Architectures.**   Furthermore, we thoroughly evaluate the performance of DCD across a range of popular model architectures, including ResNest, ResNet, ResNext, and DenseNet. The results, as shown in Table 8, highlight the robustness of our method across different choices of architectures used during model training. Notably, DCD consistently reduces the attack accuracy to 0% across all models, even when the initial attack accuracy is as high as 96%. As a data-centric defense, DCD does not require access to training procedures or the choice of model architectures. It effectively protects privacy by focusing on the data itself, ensuring that sensitive information remains secure, independent of specific modeling decisions.

Table 8: DCD's defense performance against PPA on CelebA with different model architectures.

| | ACC-all↑ | ACC-tar↑ | Att. ACC↓ | ACC-all↑ | ACC-tar↑ | Att. ACC↓ |
|---|---|---|---|---|---|---|
| | ResNeSt-101 | | | ResNet-152 | | |
| No Protection | 88.42 | 84.37 | 90.40 | 84.82 | 80.00 | 76.67 |
| DCD | 88.05 | 81.88 | 1.00 | 85.33 | 86.67 | 4.00 |
| | DenseNet-169 | | | ResNext-101 | | |
| No Protection | 84.85 | 60.00 | 73.67 | 85.89 | 73.33 | 84.67 |
| DCD | 84.32 | 60.00 | 3.00 | 87.16 | 60.00 | 2.00 |

**Generalization to Different Datasets.** We evaluate the performance of DCD using the latest model inversion attack, PPA, across multiple datasets. Table 9 demonstrates the effectiveness of DCD across different datasets, including popular face datasets such as CelebA and FaceScrub, as well as the Stanford Dogs dataset. Additionally, for each face dataset, we evaluate two GANs that have been pretrained on distinct public datasets, representing varying attack strengths. Notably, the GAN pretrained on FFHQ, which is closer to the distribution of CelebA compared to MetFaces, achieves a higher attack accuracy of 90% on the CelebA-trained model without any protection. However, our method successfully reduces the attack accuracy to 1%, highlighting its efficacy against attacks with varying strengths.

Table 9: DCD's defense performance against PPA on different datasets. The top row gives the dataset for training target models, and the second row gives the public dataset on which GAN is trained.

| | CelebA | | | | FaceScrub | | | | St.Dogs | | |
|---|---|---|---|---|---|---|---|---|---|---|---|
| | ACC-all↑ | ACC-tar↑ | FFHQ | MetFaces | ACC-all↑ | ACC-tar↑ | FFHQ | MetFaces | ACC-all↑ | ACC-tar↑ | FFHQ |
| | | | Att.ACC↓ | Att.ACC↓ | | | Att.ACC↓ | Att.ACC↓ | | | Att.ACC↓ |
| No Protection | 88.42 | 84.37 | 90.40 | 59.33 | 95.78 | 97.50 | 82.40 | 53.20 | 74.15 | 82.27 | 99.60 |
| DCD | 88.05 | 81.88 | 1.00 | 0.02 | 94.93 | 90.37 | 1.20 | 4.20 | 74.12 | 85.71 | 0.00 |

**Performance of DCD on Other Black-box MI attacks.** We extend the evaluation of DCD to include a recent black-box MI attack called BREP-MI Kahla et al. (2022). The evaluation involves two distinct model architectures applied to the CelebA dataset, face.evolve and IR152. We randomly select 6 targets, and for each target, we use BREP-MI to generate 5 samples. The results presented in Table 10 demonstrate that DCD achieving a remarkable reduction in attack accuracy to 0 for both the IR152 and face.evolve models.

Table 10: DCD's defense performance against a recent black-box MI attack, BREP-MI.

| | FaceNet64 | | | IR152 | | |
|---|---|---|---|---|---|---|
| | ACC-all↑ | ACC-tar↑ | Att.ACC↓ | ACC-all↑ | ACC-tar↑ | Att.ACC↓ |
| No Protection | 86.78 | 93.33 | 83.33 | 89.05 | 81.87 | 66.67 |
| DCD | 85.72 | 85.86 | 0.00 | 92.31 | 86.67 | 0.00 |

**Sensitive Analysis of DCD on Different Number of Protected Targets.** While baseline approaches provide binary privacy protection—either complete or none—our real-world motivation drives us to assess defense performance in scenarios where only a minority is deeply concerned about privacy. As demonstrated in the main paper, DCD offers significant advantages over model-centric baselines under such setting. We then conduct a sensitivity analysis to further explore DCD's capabilities in protecting a large portion of target classes.

Specifically, the target classifiers are trained on 1,000 identities from CelebA with the most number of samples. Surrogates samples are randomly selected from the remaining identities. We vary the number of targets for protection (i.e., 10, 500, 1000) and evaluated the defense performance of all methods against the GMI attack, a standard MI attack.

As depicted in Figure 5, DCD consistently achieved the lowest attack accuracy and demonstrated a significant advantage in preserving model utilities, even when protecting 500 of the target identities. In contrast, model-

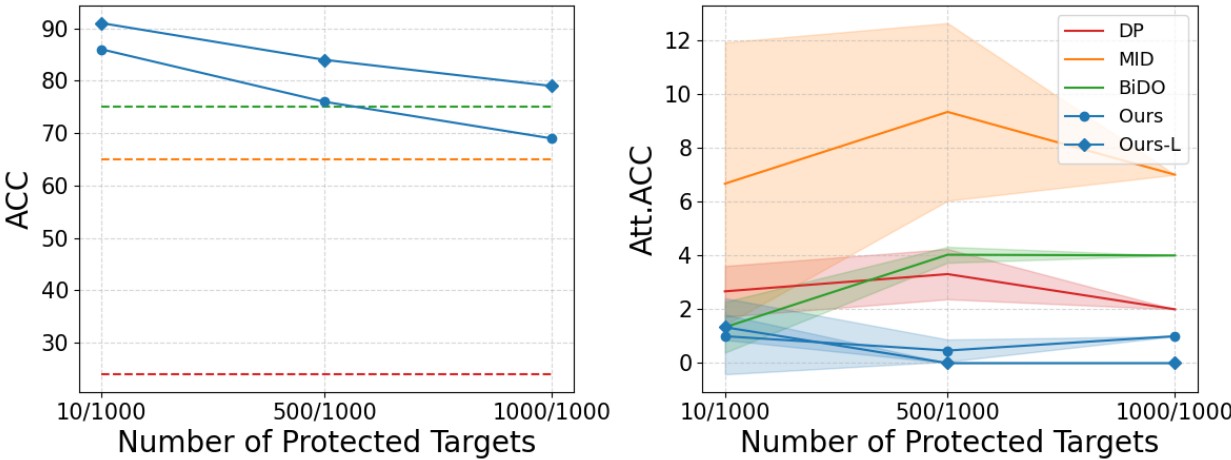

Figure 5: Defense performance against GMI on CelebA dataset. Ours-L denotes the use of DCD with a larger model (i.e., ResNet-152), whereas Ours, MID, BiDO, and DP are trained with VGG-16. The attack results are averaged over three runs, each with randomly selected protected targets.

centric baselines exhibited higher variance in attack accuracy when protecting different targets. In the case of safeguarding all of the training targets, DCD's accuracy was only slightly lower than the most advanced model-centric defense method, BiDO. This marginal difference could potentially be addressed by adopting a larger model capacity - indicated as Ours-L in Figure 5, which represents our method with a larger model (i.e., IR-152). This leads to the highest accuracy compared to all other baselines, with the attack accuracy remaining consistently low, below 1%. Implementing a larger model is also a practical option when using DCD. In practical terms, service providers adopting our strategy can judiciously select models, gravitating towards larger architectures that exhibit heightened resilience to the label noise introduced by our defense. Notably, with the amplification in model size, DP, MID and BiDO suffer a larger privacy-utility tradeoff. Consequently, they lack the leverage to utilize increased model dimensions for attenuating this tradeoff, a feat achievable by our data-centric methods.

## D.1 Evaluation against adaptive attacks

In our main setup, we have assessed the efficacy of DCD against various model inversion attacks. Anticipating that attackers could adjust their strategies if they know about the defenses, we devised an adaptive attack pipeline to test DCD further.

A simple and direct design could be recovering the original model from the one protected by DCD. To the best of our knowledge, the only prior work that obtains a clean model from the 'poisoned' version in a training-data-free fashion is Chen et al. (2023), which first applies model inversion attacks to reconstruct the training samples, then utilizes these samples to fine-tune the poisoned model. Based on this, we propose an adaptive attack pipeline as below:

1. Reconstruct training data from the model via model inversion attacks.

2. Fine-tune the model with the reconstructed data.

3. Apply model inversion on the fine-tuned model.

We ran the above pipeline on a VGG16 model in which the first 300 identities are protected using our method (ACC 80% Att.ACC 1%). And we inverted 20 samples for each protected target. GMI attack accuracy on model finetuned on these samples remained low as 1%, as the inverted samples are more resemble the surrogate sample, rather than the true targets.

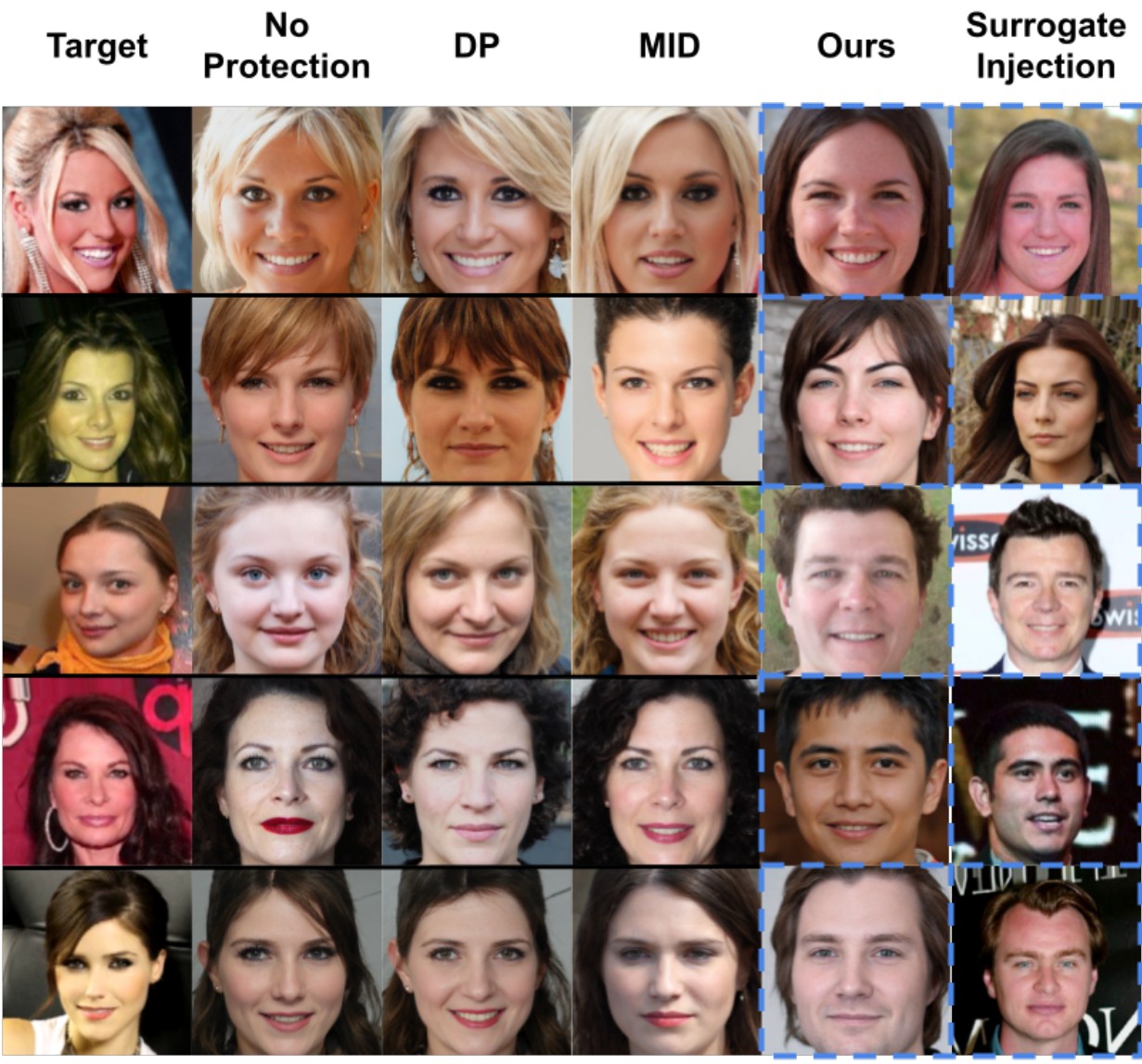

Figure 6: Visual comparison of PPA recovered samples recovered from a face recognition model trained on CelebA with different defenses. The first column displays true images for target identities. The second to fourth columns show baseline results obtained when the target model lacks protection, protected by DP and MID techniques, respectively. The fifth and final columns present reconstructions under our protection, along with corresponding injected samples. Our method successfully misleads PPA to generate samples that resemble the injected samples.

Another potential adaptive attack strategy involves optimizing for points that achieve low, but not excessively low, training loss. However, this approach presents significant challenges. It can produce numerous possible results, creating a large set of potential target samples. As illustrated in Figure 7, while the true target is a female with black hair, recovered samples that don't yield the lowest loss can appear markedly different, more closely resembling the attributes of the injected surrogates. Moreover, it would be extremely difficult for an attacker to differentiate between true targets, surrogates, and irrelevant data points within this set. This ambiguity further complicates the attacker's task of identifying the actual target samples.

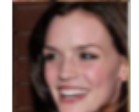 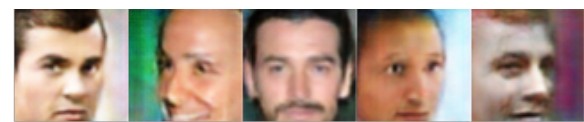

True Target    Recovered Images Not Yielding the Lowest Loss

Figure 7: Samples recovered during the MI attack optimization which not yielding the lowest loss. The leftmost image represents the true target.

