# OpenReview forum: "Data-Centric Defense: Shaping Loss Landscape with Augmentations to Counter Model Inversion"
_TMLR — Accepted by TMLR_

### Review · Reviewer_jqQz · 2024-06-04

**Summary Of Contributions:**

This paper contributes a novel approach (DCD) to protecting data privacy in machine learning models.  Previous approaches have focused mainly on techniques that can be used on the model-side, while the proposed approach is something that is done by the data owners.  The approach allows individual users to have control over the privacy of their data, while still enabling the machine learning model to perform effectively.
In addition to the proposed approach, the authors also provide theoretical justification for the success of the proposed technique, showing the impact of the technique on the loss landscape of the machine learning model.  The proposed methods is thoroughly evaluated on multiple baseline datasets and compared against existing baseline models.  It is shown to achieve superior performance when compared to these previous methods.

**Audience:**

Yes

**Claims And Evidence:**

Yes

**Requested Changes:**

None

**Strengths And Weaknesses:**

- Data privacy in machine learning is an important topic to address, especially in todays AI landscape
- The proposed data-centric defense constitutes an important approach to this problem, which offers advantages in how it can allow for individualized control over privacy decisions.
- I found the writing and organization of the paper to be very clear and understandable.
- The fact that the paper explores the theoretical foundation of the proposed approach.
- The experimental exploration is quite thorough. I appreciated the ablation study, the multiple datasets used, and the sensitivity analysis.
- Section 6 gave an appropriate ending to the paper, being clear about some of the possible shortcomings of the current work and describing challenges yet to be overcome with this data-centric approach.

Minor/Typos
- Section 4 - second sentence - "but yield the best ..." should perhaps be "but yields the best"
- Figure 3 - the units on the x-axis are not labeled.  I would suggest something like ACC (%)
- Page 11 - The heading of B. towards the bottom of the page is split across two lines unnecessarily
- Page 11 - the next sentence - "collect surrogates which ..." perhaps should be "collect surrogates for which ..."

---

> ### Author Response · Authors · 2024-09-22
> **Typos / Figure labels/ headings etc**
>
> We appreciate these comments and have modified these typos/ figures in our revised manuscript with highlights.

---

### Review · Reviewer_gKZ9 · 2024-09-04

**Summary Of Contributions:**

This paper presents a data-centric method for preventing model inversion (MI) attacks, which represent a privacy risk via reconstructing training data from trained models. ​In particular, the authors propose privacy-focused data augmentations designed to shape the loss landscape that cause MI strategies to recover irrelevant samples. This is in contrast with standard augmentations data augmentation techniques which are primarily designed to improve model generalization. Theoretical justification is provided which show the proposed augmentations reshape the loss landscape near the target and inject irrelevant samples. ​ The authors then perform a comprehensive set of empirical evaluations, comparing to a number of alternative approaches, showing performance surpasses state-of-the-art on the chosen datasets.  ​

**Audience:**

Yes

**Claims And Evidence:**

Yes

**Requested Changes:**

Please see above in weaknesses. My request for changes are largely around clarifying portions of the writing.

**Strengths And Weaknesses:**

Strengths:
1. This paper offers an interesting, and to my eyes, very novel approach to preventing model inversion attacks via data augmentation. The proposed approach is both intuitive and simple to implement.
2. The authors provide some theoretical characterizations of the proposed approach.
3. The empirical evaluation section is nice, the authors do a good job of comparing performance and performing ablation studies.

Weaknesses:
1. My main question with this paper is regarding the selection of surrogate samples. The authors provide some guidelines, but it would seem that this could possibly skew the behavior of the algorithm.
2. It also wasn't immediately clear to me what assumptions are being made about the underlying function, estimator, and attack. It would be helpful if this could be made more plain within the text.
3. The writing is a little rough in places, in particular, the introduction could use an editing pass for clarity.

---

> ### Author Response · Authors · 2024-09-22
> **The guidelines of surrogate selection could possibly skew the behavior of the algorithm?**
>
> **Re:** We appreciate the reviewer's insightful question regarding the selection of surrogate samples. To address this concern comprehensively, we'd like to clarify the impact of surrogate selection on three key aspects:
>
> - **Defense Algorithm**:
> The surrogate selection guidelines we provide do not skew the behavior of our defense mechanism. These selection choices are complementary to, rather than in conflict with, our core optimization goal of creating lower-loss, flatter-landscape surrogate injections. As demonstrated in our main evaluation (Table 1), even with random surrogate selection, DCD consistently reduces attack efficacy across various attack settings.
> - **MI Attack Algorithm**:
> Our proposed surrogate selection strategies are designed to enhance the defense effect during MI attack optimization. For instance:
> Selecting surrogates that are less similar to the target can help protect privacy at the attribute level.
> Choosing diverse surrogates can help create a smoother loss landscape around the surrogate samples, making them easier to recover compared to the true targets.
>
> - **Target Model Training Algorithm**:
> The primary optimization goal of the target model training remains unchanged: to minimize the classification loss for target images. As evidenced by Tables 1, 4, and 5, while achieving strong defense performance, DCD maintains the model's clean accuracy better than baseline methods. This demonstrates that our surrogate selection strategies do not significantly skew the behavior of the target model training algorithm or compromise its primary objective.
>
> In summary, while we provide guidelines for surrogate selection to potentially enhance performance, these do not fundamentally alter the behavior of our defense mechanism, the MI attack algorithms, or the target model training process. Instead, they offer ways to optimize within the existing framework of these algorithms. **We have added these clarifications in our revised manuscript.**

---

> ### Author Response · Authors · 2024-09-22
> **It also wasn't immediately clear to me what assumptions are being made about the underlying function, estimator, and attack. It would be helpful if this could be made more plain within the text.**
>
> **Re:** Thanks for your feedback. **The primary assumption our method relies on is the fundamental optimization goal of MI attacks**: to find an input x that maximizes the likelihood (or minimizes the loss) for a target class: $x_\text{syn}\in argmin_x L(f_\theta(x),y)$, where $y$ is the target class the attacker aims to recover, and $f_\theta$ is the victim model. **This assumption is not overly restrictive, as it forms the core principle underlying all existing MI attacks [1,2,3,4,5,6]**. While different attack methods may introduce variations in loss functions or incorporate additional constraints, they all fundamentally aim to optimize this objective. Consequently, our defense remains effective against a wide range of current and potential future attack strategies.
>
> **Beyond this core assumption, our method is designed to be broadly applicable with minimal additional assumptions:**
> - We don't make specific assumptions about the victim model architectures.
> - We don't assume any particular level of attacker access to the model (e.g., white-box, black-box).
> - We don't rely on assumptions about specific attack designs or algorithms.
>
>
>
> **In light of this feedback, we have added a paragraph in Section 3 to explicitly outline the assumptions.**
>
> [1] Zhang, Yuheng, et al. "The secret revealer: Generative model-inversion attacks against deep neural networks." Proceedings of the IEEE/CVF conference on computer vision and pattern recognition. 2020.\
> [2] An, Shengwei, et al. "Mirror: Model inversion for deep learning network with high fidelity." Proceedings of the 29th Network and Distributed System Security Symposium. 2022.\
> [3] Chen, Si, et al. "Knowledge-enriched distributional model inversion attacks." Proceedings of the IEEE/CVF international conference on computer vision. 2021.\
> [4] Struppek, Lukas, et al. "Plug & play attacks: Towards robust and flexible model inversion attacks." arXiv preprint arXiv:2201.12179 (2022).\
> [5] Kahla, Mostafa, et al. "Label-only model inversion attacks via boundary repulsion." Proceedings of the IEEE/CVF conference on computer vision and pattern recognition. 2022.\
> [6] Yuan, Xiaojian, et al. "Pseudo label-guided model inversion attack via conditional generative adversarial network." Proceedings of the AAAI Conference on Artificial Intelligence. Vol. 37. No. 3. 2023.

---

> ### Author Response · Authors · 2024-09-22
> **Improve the writing for clarity.**
>
> Thank you for your suggestion! We have updated our manuscript based on your feedback.

---

### Review · Reviewer_jUiK · 2024-09-09

**Summary Of Contributions:**

The authors propose a method for reducing the risk of model inversion, i.e. to make it harder for attackers with access to a model to reconstruct its training dataset. The authors first notice that most model inversion attacks work by optimizing (in a regularized space) for points that minimize the training loss. The proposed defense is to introduce irrelevant points that have even lower training loss than the actual points of interest, hoping that the attackers will recover the irrelevant points instead of the true ones. The authors show experimentally that their method decreases the proximity of the recovered samples to the training ones across different architectures and datasets.

**Audience:**

Yes

**Broader Impact Concerns:**

I believe that the authors should clearly state that there is no guarantee that their method won't leak training samples. A better attack in the future could break the proposed method.

**Claims And Evidence:**

Yes

**Requested Changes:**

* Do the surrogate samples have to be meaningful? What if random noise is used to form these surrogate samples? This could mitigate some of the concerns listed in the weaknesses section.

**Strengths And Weaknesses:**

Strengths:
* The paper is well-written and well-motivated. I really enjoyed reading the introduction section, which sets the scene for what is about to follow. The authors motivate their work by saying that the privacy needs of different users are different. This can be leveraged to develop privacy-protecting techniques that operate on a sample level. The proposed method belongs to this category since it allows selecting the training augmentation/corruption for each sample/category.
* The topic is timely and the direction of introducing changes to the data, instead of changes to the algorithm, is promising.
* The algorithmic idea behind the paper is clever. If the attackers optimize for the lowest error, they will end up recovering irrelevant points.
* The theoretical analysis and the proofs seem correct.
* The experiments show significant improvements over prior work. In some cases, the authors manage to almost maintain the performance of the models trained without any protection, with attacker accuracy being 0.

Weaknesses:
* I am not sure about the potential unintentional outcomes of the proposed method. Models trained with these methods would be fooled by the surrogate samples. This can have very unpredictable and negative outcomes when these models are used in real systems, e.g. for face recognition as the authors propose.
* The proposed method is very specific to the way current attacks for model inversion work. It is not clear whether small changes to these attacks would lead to recovery of the samples used in the training. For example, the attacker might optimize for points that achieve low training loss, but not too low or something along this intuition.
* It is unclear how the method scales as the number of classes grows. For the majority of the experiments, the number of classes seem to be pretty low.
* The authors have missed some relevant works on this topic. The idea of introducing corruption to decrease memorization is not new. Particularly, there is a line of works in the context of generative models that studies this. In generative models, the decrease of memorization leads to an increase in diversity which sometimes leads to improved FID scores. There have been a few works that have been exploring this, either for performance reasons or to reduce copyright/privacy concerns. I believe that the paper would greatly benefit from discussing these connections:

[1] Elucidating the Design Space of Diffusion-Based Generative Models
[2] Ambient Diffusion: Learning Clean Distributions from Corrupted Data
[3] Consistent Diffusion Meets Tweedie: Training Exact Ambient Diffusion Models with Noisy Data
[4] Be like a Goldfish, Don't Memorize! Mitigating Memorization in Generative LLMs
[5] Understanding and Mitigating Copying in Diffusion Models

---

> ### Author Response · Authors · 2024-09-22
> **The potential unintentional outcomes of the proposed method**
>
> **Re:** We recognize and appreciate your concerns about the unintentional outcomes related to the use of surrogate samples, particularly the recognition of surrogate samples by the trained models.However, we would like to clarify that this classification is crucial for a successful defense, allowing surrogate samples—rather than authentic target samples—to be retrieved when the model is under attack.  We have also addressed potential outcomes below, explaining why our defense strategy doesn't introduce additional risks:
>
> - **Potential for attackers to bypass the classification system using recovered surrogates**: It's important to note that even without surrogate training, a model inversion attack could still recover samples categorized as the target class. The primary security risk stems from the model inversion attack itself. Our defense operates within this existing threat landscape and doesn't introduce new vulnerabilities beyond those inherent to the inversion attack.
> - **Risk of system bypass by individuals with access to surrogates**: Users aiming to protect their privacy have discretion over surrogate selection. As there's no incentive for users to disclose their surrogate choices, it's unlikely for an adversary to identify a user's specific surrogate without resorting to a model inversion attack.
>
> In conclusion, our method is designed to work within the existing threat model of model inversion attacks, providing valuable privacy protection without significantly increasing vulnerabilities. For a more comprehensive discussion, please refer to Section 6.

---

> ### Author Response · Authors · 2024-09-22
> **The proposed method is very specific to the way current attacks for model inversion work ... the attacker might optimize for points that achieve low training loss, but not too low or something along this intuition.**
>
> **Re:** We appreciate the reviewer's insightful consideration of potential adaptive MI attacks. We'd like to address this concern by emphasizing two key points:
>
> - **Fundamental Principle of MI Attacks**:
> Our proposed method is grounded in the core idea underlying all existing MI attacks [1,2,3,4,5,6]: the true target images should achieve the highest log-likelihood under the target model. **This principle is consistently utilized across the spectrum of MI attack research**, despite variations in specific loss functions or model designs. By targeting this fundamental aspect, our defense mechanism maintains its effectiveness against a wide range of attack strategies.
> - **The adaptive attacks can be hard**:
> While it's theoretically possible for an attacker to modify their strategy, such as optimizing for points with low but not too low training loss, such an optimization could yield numerous possible results, creating a large set of potential target samples (Figure 7 in Section D.1). And it would be extremely challenging for the attacker to distinguish which samples in this set are true targets and which are surrogates or irrelevant data points. This difficulty arises precisely because the attacker's original goal is to identify the target samples.
>
> [1] Zhang, Yuheng, et al. "The secret revealer: Generative model-inversion attacks against deep neural networks." Proceedings of the IEEE/CVF conference on computer vision and pattern recognition. 2020. \
> [2] An, Shengwei, et al. "Mirror: Model inversion for deep learning network with high fidelity." Proceedings of the 29th Network and Distributed System Security Symposium. 2022.\
> [3] Chen, Si, et al. "Knowledge-enriched distributional model inversion attacks." Proceedings of the IEEE/CVF international conference on computer vision. 2021.\
> [4] Struppek, Lukas, et al. "Plug & play attacks: Towards robust and flexible model inversion attacks." arXiv preprint arXiv:2201.12179 (2022).\
> [5] Kahla, Mostafa, et al. "Label-only model inversion attacks via boundary repulsion." Proceedings of the IEEE/CVF conference on computer vision and pattern recognition. 2022.\
> [6] Yuan, Xiaojian, et al. "Pseudo label-guided model inversion attack via conditional generative adversarial network." Proceedings of the AAAI Conference on Artificial Intelligence. Vol. 37. No. 3. 2023.

---

> ### Author Response · Authors · 2024-09-22
> **It is unclear how the method scales as the number of classes grows. For the majority of the experiments, the number of classes seem to be pretty low.**
>
> **Re:** Thanks for raising your concerns. **In Section 4.2, we conducted a sensitivity analysis on the number of classes to be protected for a face recognition model trained on the CelebA dataset against GMI attack**. We vary the number of targets for protection (i.e., 10, 500, 1000) and evaluate the defense performance of all defense methods. As depicted in Figure 3, DCD consistently achieved the lowest attack accuracy and demonstrated a significant advantage in preserving model utilities. In the case of safeguarding all 1000 training targets, DCD’s accuracy was only slightly lower than the most advanced model-centric defense method, BiDO. This marginal difference could potentially be addressed by adopting a larger model capacity - indicated as Ours-L in Figure 3, which represents our method with a larger model (i.e., IR-152). This leads to the highest accuracy compared to all other baselines, with the attack accuracy remaining consistently low, below 2%.
>
> We would like to emphasize that as stated in the introduction, **our research is driven by the quest for personalized privacy controls**. On one side, according to extensive social science research (e.g., see a recent survey conducted by Cisco [7], **only a relatively small fraction of users rank privacy leakage as a high concern**. Conversely, recent regulations are increasingly advocating for individualized control. Yet, prevailing model-centric defenses offer a **binary** approach to protection: either all are safeguarded or none are. This type of approach is unable to optimize utility even when there's a limited number of privacy-concerned individuals in the population. Our motivation, rooted in real-world relevance, prompts us to examine defense performance in scenarios where **only a minority** is deeply concerned about privacy. As such, we randomly picked a small proportion of users as the target and evaluated the attack/defense performance on these users.
>
>
> [7] Cisco. Cisco 2022 Consumer Privacy Survey, 2022.

---

> ### Author Response · Authors · 2024-09-22
> **Discussion on relevant work that introducing corruption to reduce memorization for generative models.**
>
> **Re:** Thanks for your feedback! While we acknowledge the importance of the works mentioned, we'd like to clarify the key differences between our approach and the cited studies:
>
> - **Focus and Objective**: Our work specifically targets classification models and aims to reduce the risk of model inversion attacks. In contrast, the cited works primarily focus on generative models and aim to reduce memorization for improved diversity or performance. The fundamental goals and challenges in these two domains are distinct, leading to different approaches and outcomes.
> - **Methodology**: Our method introduces novel, carefully designed augmentations to the training data, rather than simply injecting noise. This is a crucial distinction, as prior research has shown that directly adding noise to training samples can potentially make classification models more vulnerable to model inversion attacks [8]. Our augmentations are specifically tailored to shape the model's loss landscape in ways that thwart model inversion attempts while maintaining classification accuracy.
>
> In response to your comment, we have incorporated these discussions in our revised manuscript.
>
> [8] Mejia, Felipe A., et al. "Robust or private? adversarial training makes models more vulnerable to privacy attacks." arXiv preprint arXiv:1906.06449 (2019).

---

> ### Author Response · Authors · 2024-09-22
> **Do the surrogate samples have to be meaningful? What if random noise is used to form these surrogate samples?**
>
> **Re:** The use of meaningful surrogate samples, as opposed to random noise, is indeed a crucial aspect of our approach: **The primary goal of our method is to fool the attacker, making them believe that the recovered images belong to the true target when they actually do not.** Using meaningful surrogate samples that belong to similar distributions as the private targets (e.g., using a different public identity's face image to protect a face image) is more effective in achieving this goal. Random noise would be easily distinguishable from real data, potentially alerting the attacker to the presence of a defense mechanism. Moreover, sophisticated attackers might employ additional filtering or post-processing techniques to discard unrealistic outputs. By using samples from similar distributions, we ensure that the recovered images remain within the realm of plausible data, making the defense more robust against advanced attack strategies. Moreover, many state-of-the-art MI attacks employ GANs to regularize the optimization process [1,2,3,4,5,6]. **These GAN-based methods are specifically designed to generate realistic samples from the target distribution, which makes it extremely challenging to reconstruct random noise patterns**, as such patterns lie far outside the learned data distribution.

---

### Comment · Reviewer_jqQz · 2024-10-03

I have read the author's responses and appreciate the thought and care they took in addressing the comments and questions of the reviewers.

---

### Decision · Action_Editor_2vFV · 2024-11-06

**Recommendation:** Accept as is

**Comment:**

My comments are incorporated in "claims and evidence."

**Audience:**

The manuscript presents a defensive mechanism to reduce the attack success of model inversion attacks via flattening the loss landscape. The connection between the loss landscape and the MI attack success has not been well-explored, so the audience concerned about their ML models' privacy risks will be interested. But, I also resonate with one of the reviewers who pointed out that the defense is tailored for a certain MI attack type.

**Claims And Evidence:**

The manuscript has been significantly improved through the author-reviewer discussion. The authors put a lot of effort into polishing the introduction, adding relevant works, clarifying threat models, and discussing potential limitations. My assessment is that the empirical evaluations back most claims.